# The Relationship between Birth Order, Sex, Home Scholarly Culture and Youths' Reading Practices in Promoting Lifelong Learning for Sustainable Development in Vietnam

**Trung Tran** [1], **Thi-Thu-Hien Le** [2], **Thu-Trang Nguyen** [1], **Anh-Giang Pham** [3], **Thi-Hanh Vu** [4], **Minh-Hoang Nguyen** [5], **Ha-My Vuong** [6], **Thu-Trang Vuong** [7,*], **Phuong-Hanh Hoang** [8], **Manh-Toan Ho** [9,10,*] and **Quan-Hoang Vuong** [9,10,11,*]

1   Vietnam Academy for Ethnic Minorities, Hanoi 100000, Vietnam
2   Faculty of Pedagogy, University of Education, Vietnam National University, Hanoi 100000, Vietnam
3   Faculty of Natural Sciences, Hongduc University, Thanh Hoa 40000, Vietnam
4   School of Economics and International Business, Foreign Trade University, Hanoi 100000, Vietnam
5   International Cooperation Policy, Graduate School of Asia Pacific Studies,
    Ritsumeikan Asia Pacific University, Beppu, Oita 874-8577, Japan
6   Hanoi Amsterdam High School for the Gifted, Hoang Minh Giam Street, Cau Giay District,
    Hanoi 100000, Vietnam
7   Sciences Po Paris, 75337 Paris, France
8   National Centre for Sustainable Development of General Education Quality, Vietnam National Institute of
    Educational Sciences, 101 Tran Hung Dao Street, Hoan Kiem District, Hanoi 100000, Vietnam
9   Center for Interdisciplinary Social Research, Phenikaa University, Ha Dong District, Hanoi 100803, Vietnam
10  Faculty of Economics and Finance, Phenikaa University, Ha Dong District, Hanoi 100803, Vietnam
11  Centre Emile Bernheim, Université Libre de Bruxelles, 1050 Bruxelles, Belgium
*   Correspondence: thutrang.vuong@sciencespo.fr (T.-T.V.); toan.homanh@phenikaa-uni.edu.vn (M.-T.H.);
    hoang.vuongquan@phenikaa-uni.edu.vn (Q.-H.V.)

**Abstract:** Book reading is an important factor contributing to children's cognitive development and education for sustainable development. However, in a developing country like Vietnam, statistics have reported a low figure in book reading: only 1.2 books a year. This research study used a dataset of 1676 observations of junior high school students from Northern Vietnam to explore students' reading behavior and its association with demographic factors, and the family's reading culture. Data analysis suggests the older the student gets, the less inclined they are to read, and being female and having hobbies of low sensory stimulation are linked to higher preference for reading. Regarding scholarly culture at home, students who read more varied types of books and spend more time on books are correlated with higher reading interest. Reading habits are also positively reinforced by the capacity to access books and parental book reading.

**Keywords:** reading practices; reading abilities; junior high school students; sustainable education; education for sustainable developments; gender; Vietnam

## 1. Introduction

*1.1. A Brief Overview*

> *"You don't have to burn books to destroy a culture. Just get people to stop reading them"*
>
> —Ray Bradbury

Humankind has always concerned itself with preserving what they have learned and created and passing it down to the next generations. As the printing press facilitated the production of written materials, oral traditions gave way to books as the dominant medium of recording human knowledge. Even cultures that to this day rely more on word-of-mouth would perhaps benefit from being documented in written forms. Reading books, though not the only means, is therefore crucial in the process of gaining knowledge (see Figure 1). Maintaining appreciation for book reading has become essential in nurturing and sustaining cultures.

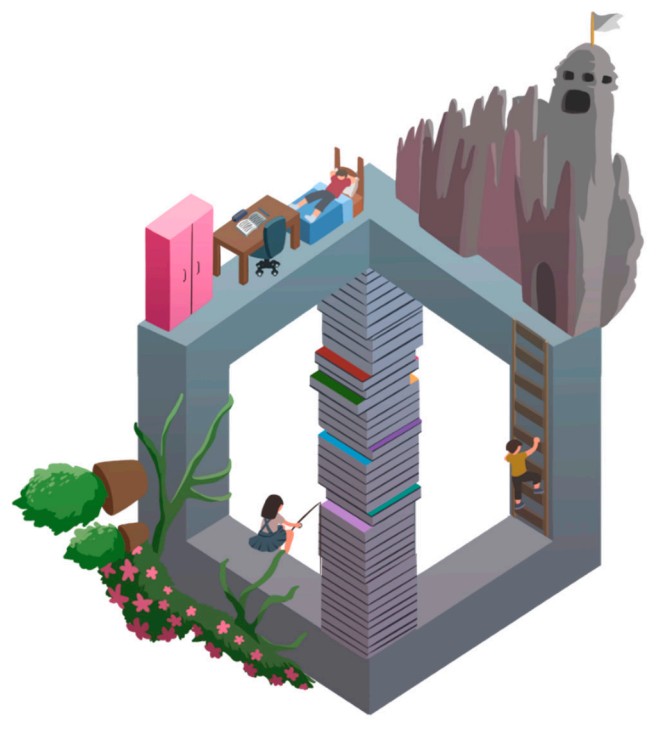

©HA-MY VUONG

**Figure 1.** Readers give it a caption! (© 2019 Illustration courtesy of Ha-My Vuong).

Reading is the fundamental process of learning that helps stimulate social awareness and critical reflection. In this sense, building a literate and learning society is prerequisite to innovation initiatives and social change. Literacy achievement is, therefore, widely recognized as a key indicator of a developed nation. Particularly, in the Sustainable Development Goals (SDGs), the United Nations has considered the acquisition of necessary knowledge and skills for sustainable development as the target 4.7 of the goal for Quality Education [1]. Since the ultimate goal of education in today's world is no longer to transmit knowledge and information but to foster learner autonomy and lifelong learning, reading, as one of the most basic methods to expand knowledge and understanding, is of great concern. Research has confirmed the crucial role of reading that expands beyond enhanced literacy outcomes to various cognitive capabilities [2]. In addition, reading as early academic skills has also been linked with better academic achievement, educational and occupational attainment in young adulthood [3,4]. Learners' interest and proficiency in reading as well as their reading practices are one of the main focuses of a modern and sustainable education system.

As today's technology-driven world is shifting toward a new era of computational power, and computational entrepreneurship [5], literacy is also emphasized as a key skill in aiding 21st century leaners' acquisition of digitized information. When examining the relationship between scholarly culture from childhood and adulthood information and communication technology (ICT) skills, Evans, Kelley, Sikora and Treiman [4] found that developing a reading culture at home is associated with better techonological problem solving skills. The benefits of reading have also been confirmed in Sciences, Technology, Engineering and Mathematics (STEM) education domains since the practice of

reading helps formulate new ideas and inquiries for the problem solving processes involved in STEM learning [6]. As a result, youth's reading practices are among one of the key focuses of education in order to develop human capital, lift labor productivity and promote sustainable development in the age of information.

However, in developing countries like Vietnam, sustainability is difficult to achieve due to a lack of knowledge, skills, and awareness of global issues in ordinary citizens. This partly results from the absence of a promoted reading culture in the country. According to statistics from Dammio.com [7], 70 million Vietnamese people own a mobile phone, and around 64 million people are Internet users. An average Vietnamese spends up to 11 h a day for Internet, social media, and consuming contents, which include TV, video, and music. Vietnamese people have, in fact, become more and more comfortable with the omnipresence of technology in their life, so much so that even old habits seem to be gradually replaced. Statistics from Vietnam Ministry of Education and Training and Room to Read reported a Vietnamese person reads only 1.2 books per year [8]; including textbooks, the number rose up to four books per year, according to the Vietnam Publishers Association [9]. Strangely, the publishing industry was having a good year with around 20,000 titles in 2018; moreover, a bookstore system also had a 15–20% rise in revenue [9]. In education, reading culture does not seem to affect the achievement of Vietnamese students, especially in science and mathematics. The evaluation of Vietnamese students via the Programme for International Student Assessment (PISA) test shows high scores in these two areas. In the Science test, Vietnam was ranked 7th out of 65 countries in 2012 and 8th out of 72 countries in 2015. In the Mathematics test, Vietnam scored better than the international average in both years. However, when it comes to Reading, Vietnam declined from rank 15 in 2012 (Score 511) to rank 32 in 2015 (Score 495). A comprehensive understanding of the influencing factors of reading practices is substantial for fostering sound and scientific policies that will result in educational sustainability.

According to a recent study, reading interest and high school students are not well researched by researchers in Vietnam: only 19 out of 174 articles of Vietnamese researchers that were indexed in Web of Sciences are about general education [10]. Thus, based on a comprehensive dataset of 1676 records of Vietnamese junior high school students, students' demographic information and family background were taken into consideration with their book reading habits. This research aims to address the need to study the reading habits of Vietnamese youths in order to promote lifelong learning skills, provide grounded evidence for educational policy in Vietnam [11,12], and improve equity in education and science [13–15].

### 1.2. Literature Review

There have been many attempts at conceptualizing the act of reading, evidenced by the number of fragmented reading theories in cognitive science [16]. The works of Carver theorized reading and comprehension based on generalizations on thought communication and coined the notion of "rauding" as a more general process of language comprehension, of which reading was a special case [17]. The simple view of reading defined the process of reading as a product of decoding (as in word recognition) and linguistic comprehension [18,19]. According to this approach, the act of reading could not be achieved with either component missing, and when they are both present, they have equal potentials of contributing to the output of the reading process. In more concrete terms, it means that knowing a language (linguistic comprehension) does not guarantee literacy in said language: for example, children who have not learned the alphabet nor how to spell would not be able to decode written words, therefore unable to read. Vice versa, knowing an alphabet or even the orthography of a language (decoding skill) does not mean one could understand a text written in said language, without the necessary comprehension skills (vocabulary, grammar, etc.).

Two main problematic aspects arise from the usage of decoding and language comprehension as the main components to define reading. On the one hand, the purest measure of decoding relates to spelling-sound correspondence, which would make it complicated to define literacy in the case of disability. On the other hand, this definition puts language on center stage and begs the question as to

how research on reading ability in various languages around the world could have a coherent common frame, especially in terms of measuring linguistic comprehension. In fact, it could be noted that while there is an extensive literature on the subject of reading, the majority of them are limited to reading in the English language. Linguistic skills aside, this also implies a prevalence of Anglosphere-based studies, with all of its implicit cultural assumptions. Taking this into consideration is not only crucial in reviewing the literature, but also in assessing the contribution of our paper, which is based exclusively on the Vietnamese language. That being said, the simple view of reading remained crucial in the literature on reading and cognitive skills; most of its critics also attempted to build up from it or merge it with existing theories [20]. Ouellette and Beers, for example, still based their hypotheses on the model of the "simple view" while suggesting more complex constructs under the two main components, and the importance of oral vocabulary in reinforcing both word recognition skills and listening comprehension [21].

Regardless of the theorized models, it had consistently been proven that reading contributed positively to cognitive development and literacy skills, particularly that of children [22–24]. A large body of literature pointed to a decline in reading practices along with age in the United States, in the general adult population (aged 20 and up) [25] but also among undergraduate students [26] as well as elementary (Grade 1 through 6) pupils [27]. The same phenomenon could be observed among primary and secondary school pupils (mainly aged 6–16) in England [28]. Anderson, et al. [29] showed that secondary school students have less interest in reading than elementary students. Panel data individually following 164 middle school students (from sixth to eighth grade) in the US over a 3-year time period have indicated a decline in voluntary reading despite their purposes for reading remaining stable [30]. On the other hand, despite the development of multimedia mediums and other sources of information such as the Internet, generational shifts in reading frequency did not seem to have a negative effect on reading. In fact, longitudinal data on a large scale suggested that the generational shifts in reading frequency is not absolute and could be conditional on reading materials in terms of genres (magazine, newspaper, literature, etc.) and on the physical medium of reading (physical or digital copy). Specifically, Robinson [25] addressed this matter with an optimistic outlook for reading as an equal to other media (television, for instance) in terms of consumption, while specifying the shift in reading material preference—a decline in newspaper and increase in books and magazines—in the American adult population. Shahriza Abdul Karim and Hasan [31] studied a sample of Malaysian college/university students (aged 19–22) and pointed out the fact that students maintained their reading practices, and only changed their media from physical books to websites. While this observable shift in reading media was considered inconsequential for college-age students, it might not be the case for readers of all age ranges. It has been shown that the presence of electronic features in reading materials can diminish the positive effect of parent-child storybook reading on literacy skills development among children [32].

Another focus in the literature on reading practices was its relationship with regards to reading enjoyment. There has been extensive documentation on the decline of reading enjoyment with age, particularly in students. Clark and Foster [28] reported that elementary pupils both enjoyed reading more and rated their reading proficiency higher than their secondary counterparts. Ley, Schaer and Dismukes [30] pointed out the significant relationship between reading attitude and reading behavior of students, which both declined throughout junior high school regardless of gender, race, or socioeconomic status. The majority of students prioritized utilitarian values of reading, suggesting that extrinsic motivations, such as to seek certain information for a specific purpose or to complete a class assignment, were more instrumental in leading students to read. This was further confirmed by later studies in other countries, such as that of Majid and Tan on schoolchildren (aged 9–12) in Singapore [33]. However, Guthrie, et al. [34] reported the important role of intrinsic motivations as well as situational interest in certain books in increasing long-term reading practices. In addition, the motivations for reading and by extension the level of enjoyment derived from reading may vary on an individual basis in relation to personal preferences regarding the type of books [34,35]. Finally,

for college students, leisure book reading, while still valued, was less prioritized than other forms of entertainment (watching television or chatting with friends over text messages) [26]. One of the reasons was that they had already read a lot for school, which may suggest that obligatory reading could influence the level of reading enjoyment in general. Recently, researchers have discussed the lack of sciences and philosophy books for children, and Pigliucci [36] called for attention in this matter.

Gender has also been examined in relation to reading patterns and practices. Research studies in both Western and Eastern societies have shown that males and females are often reported to differ significantly in reading enjoyment, motivation, and reading material preferences [28,31,33,37,38]. Most of these results show that girls read more and are more interested in reading than boys; however, findings based on data from 12 to 15 years old Australian students suggested otherwise [39]. In terms of race, Asians were slightly better at reading reports, while Whites found reading newspapers and novels easier [40]. A recent research article showed that third grade elementary school female students demonstrated similar reading ability as their male counterparts, but they valued reading more [41].

Children's reading practices and the forming thereof often involve the home as much as they are associated with the school. Clark and Foster [28] found that students in England generally agree that both family and school should teach and motivate children to read. In fact, numerous works in the literature have pointed out the positive role of dialogic parent-child storybook reading in developing child literacy [42,43]; this applies to natives of languages other than English as well [44]. Palani [45] suggested parental guidance as one of the measures to develop reading interest. Ennemoser and Schneider [46] used parent's reading for children as a substitute for exposure to reading materials in kindergarten children. This methodological choice suggests the crucial role of parental guidance in cultivating an interest in reading for children in their early formative years, especially when they have not yet learned to read. In England, pupils also reported their parents as being their most important reading partners and source of influence in regards to reading practices [28]. But parent-child interactions were not the only element to be taken into account, regarding book reading and reading competency development in the home. In some cases, merely having access to books at home during childhood already improved cognitive ability of children [22]. Other studies have also shown that dialogic parent-child reading might not always bring the expected results of child reading achievements [47]. Regarding other attributes related to the household, Clark and Foster [28] reported that pupils who are eligible for free school meals, implying a background of lower socio-economic status, are less enthusiastic readers; their parents also read less or own fewer books. Compton-Lilly [48] had, based on longitudinal data obtained through an 8-year long qualitative study of an African-American family, gained insights into the role of familial and socioeconomic context on forming children's discourse and literacy: namely, that the language used in the family, the economic situation of the family and the manner in which family members discussed literacy contributed into shaping the child's book choices and enjoyment of school texts.

Related to family literacy is parent-child reading activities, and, by extension, parental attention and reading habits in children. In effect, the inverse relationship between family size and birth order on the one hand and childrearing quality on the other has been documented in the literature [49–53]. One of the theories most often drawn upon is the resource dilution model, relating parental resource to child quality. Assuming that parental resource—both material (access to books and other forms of education) and financial (support for college tuition, for example) as well as in terms of parent-child interaction—is limited, it would have to be divided between siblings. It follows that the larger the family size, the less parental resource each child would receive on average; this has been evidenced by numerous studies [49,50,52]. In terms of accumulated parental resource since the birth of a child, the firstborn would receive the most attention, because there would be a period of time during which they enjoy undivided attention from their parents, in the absence of siblings, while the opposite could be said for the youngest child in the family who would have been born into a comparatively lower average share of parental resource. It could then be the case that in theory, the more older siblings a child has, the less they receive from their parents relatively, whether it be the resource on average at a

point in time, or the cumulative investment throughout their formative years [54]. Evidence for this negative effect of higher birth order (meaning, having more older siblings) on children can be found in work by Black, Devereux and Salvanes [50], who have also pointed out that the negative effect applies regardless of the socioeconomic status of the household; although studies such as Steelman and Powell [53] suggested minor financial advantage later in life among younger siblings.

As has been mentioned above, alongside the development of technology was the shift in reading materials and media. For instance, web-based educational platforms are expected by the teachers to be the new way for students to learn [55]. The optimistic outlook for books as seen in Robinson [25] could, however, be put back into perspective, as other forms of information dissemination and entertainment had the potential to dwarf written mediums. Mokhtari, Reichard and Gardner [26] found that Internet activities do not seem to interfere with reading practices, whereas watching TV is a popular activity but not as enjoyed as the Internet. They also showed concerns for students' habit of "multitasking," which often combines reading with listening to music or watching TV and the efficiency of such activities. The study highlighted the lack of time in student life and overlap of activities. The subsequent question to reading practices would perhaps be reading ability. Findings in Ennemoser and Schneider [46] based on the data from 332 German children collected from 1998 to 2001, suggested that TV viewing had at least a medium-term effect on reading ability (three-year gap), especially at an early age. It should also be noted that the genre of the TV program also mattered in assessing the influence of TV consumption: more specifically, the effects of consumption of entertainment-general audience programs tend to support the inhibition theory (basically, that children read less as they watch more TV), as opposed to educational programs that might enhance reading practices. Compton-Lilly [56] further suggested that teachers can observe how video games engage children to find out good ways to teach children how to read. Multimedia consumption can thus, to an extent, be analog to reading practices and the formation of reading habits in children.

But children and adolescents do not only either read books or watch television. In fact, pastimes other than book readings have existed long before the birth of the screens, and there is a plurality of hobbies so complex that many classifications have been constructed for them, such as the Holland's RIASEC (Realistic, Investigative, Artistic, Social, Enterprising, and Conventional) model [57,58]. We have touched upon reading enjoyment, so of course, it would only be natural to look at leisure reading not only about obligatory reading but also to other potential leisure activities. The question is, which factors relating to leisure activities would be relevant in the context of determining reading practices?

Profiling leisure interests is not a simple task, especially where it concerns adolescents. While frameworks, such as the RIASEC model, have attempted to categorize leisure interests, most of these are related to professional capacity and/or opportunity, thus more fitting for adults. For adolescents, there are no dominant methods of categorizing leisure activities. Garton and Pratt [59] group 73 leisure activities into six groups, namely: Sport, Gregarious, Water sports, Serious, Indoor games, and other activities. Fitzgerald, et al. [60] devised two sets of activity categories. The first set is applicable for measuring participation and consists of Sports, Outdoors activities, Keep fit, Non-sports, Entertainment, Computer/Friends, the other for interest (Outdoors activities, Entertainment, Sports, Social activities, Hobbies, Others).

Leisure activities are highly related to personal interests, as evidenced by strong correlations between interest and level of participation in an activity [59,60]. Through the analysis of English schoolboys test results, Hudson [61] made the distinction between two personality profiles among male pupils: "convergers", who score higher on technical topics but poorly on open questions; and "divergers," who are the opposite. The paper suggests that converger-type boys are more likely to restrict themselves to impersonal, technical topics so as to avoid open-ended discussions, controversies, and matters deeply involving humans and feelings. This study, however, only provides a correlation between personality and interest rather than a relationship between these two factors; it is also only limited to male pupils. It has been found that personal interest plays an important role in determining the level of participation [59,60]. In the recent years, however, there has been evidence to suggest

that external factors and environmental change also play a role in determining activity preference among children and adolescents [62,63], in terms of both intensity and content of the activity. There is, notably, a global shift towards more passive activities such as TV watching, and away from free-play and experiential learning, regardless of the child's gender or the family's socioeconomic status.

There is evidence to suggest a stark interest-based difference of hobby choice between adolescent boys and girls. Sex is, in fact, an important predictor for the level of both interest and participation in leisure activities [59,60]. It would be impossible to draw a clear line of distinction between what constitutes strictly masculine or strictly feminine interests. However, there have been many attempts at generalizing sex-based differences in occupational and/or leisure interests between males and females, based on a general consensus that such a distinction does exist, either between organic and inorganic things. One well-known example would be Things versus People distinction, conceptualized as a dimension of Holland's RIASEC model of occupational interest, which has been employed in representing leisure [64] as well as in investigating sex-based differences in interests [65].

It has already been mentioned above that female and male reading practices differ. However, little has been said directly about the link between female and male interests in general and their practicing of reading (in terms of both content and quantity). Thus, this paper aims to bridge the gap by studying the role of sex-based differential interests as an indicating factor of reading practices.

While the literature mentioned in the above paragraphs pertain largely either to reading in the context of learning or to reading among students, it should nonetheless be noted that reading is not inherent to learning and vice versa. The goal of reading is not limited to being purely utilitarian or based on knowledge-gaining. In the conceptual framework of this survey, however, we aimed to study the activity of reading in specific relations with education and future occupational aspirations, hence the focus on measurable cognitive and competence-wise benefits of literacy. In addition, it should be noted that despite our study being anchored on print-based literacy, we are aware of the wealth of literacies beyond the exposure to and absorption of information in textual form [66]. By linking reading skills to pedagogy and the sustainable development of the education system in our country, we are by no means claiming that print literacy is the only measure of academic success or cognitive development. It is in fact a construct pertaining to the school of development, which to this day remains the dominant model yet not without facing staunch criticism [67,68]. Print-based literacy did in fact emanate from urban and industrialized societies, therefore favoring those over rural and agrarian ones. It is crucial to acknowledge this in order not to undermine communities in which other forms of literacies and wisdoms prevail, especially in the quest for and the preservation of more sustainable ways of transferring knowledge and of living [69].

### 1.3. Notions and Concepts

In this study, we intended to keep the concept of reading as encompassing and open to interpretation as possible, as the subjects of the research are junior high students from grade 6 to 9, which in most cases correspond to the age range from 11 to 15. Respondents are allowed to apply their intuitive understanding of the term "reading", with only minimal instruction in order to assure a global coherence between filled records. The activity could thus be understood as meaningful exposure to texts that are not necessarily intensive. This means that leisure reading is included and there is little limit on the reading material and subject. Comic books, for example, were not excluded, and with good reasons: we consider the ability to relate visual representations with textual information—albeit of shorter length—and to process both simultaneously just as appreciable as simple reading skills of more word-dense texts. This open understanding of reading also means that any reading medium, i.e., paper-based, digital, etc. could be included.

In addition, several recurring terms related to reading necessitate a consistent understanding. First, the term "reading interest", and denotes the student's proclaimed interest in the activity of reading. More specifically, in the original questionnaire, we posed the question "Do you like to read?". Students were given the option to answer either "yes" or "no". As all self-reported measures go,

we acknowledge that this answer might be under the effect of social desirability and would reflect with varying extents of accuracy the student's true attitude regarding the activity. For this reason, we focused on the differing behaviors and habits between students who answered "yes" and those who answered "no". This notion corresponds to the variable "Readbook".

Second, the terms "reading habits" and "reading practices", in the context of this paper, were used interchangeably and refer exclusively to the amount of time spent reading daily. It measures student behaviors regarding the activity, to an extent, and were thus used as inputs to examine how already formed habits affect the likelihood that they show an interest in reading. There is the underlying assumption that people, in their formative years, depend on outside influences to form the habit of engaging in literary activities—as has been mentioned in the literature concerning family literacy. Although this measure is also self-reported, it seems to be less under the effect of social desirability, as respondents did not hesitate to report the lowest amount of time per day spent reading either type of books. The corresponding variables are "TimeSoc" and "TimeSci", which measures the time students spend reading social sciences and humanities books and natural sciences books, respectively (see Appendix A, Table A1).

Regarding external factors relating to the students' habits, i.e., family and school, we have decided to focus on how the family environment is linked to the reading interest among students. Two types of reading encouragement were examined: passive encouragement, indicated by the provision of books by the parents; and active encouragement, indicated by parental accompaniment in the form of reading books out loud for their children. The corresponding variables are "Buybook" and "Readstory". The exact wording for the questions (translated into English) are, "Do your parents buy books for you?" and "Do your parents read you stories?", respectively. The answer "yes" denotes that the event mentioned in the question has occurred at least once; conversely, the answer "no" means that the event has never occurred.

It follows that both "Buybook" and "Readstory" are dichotomous variables, which might seem overly reductive, for an act as fluid in intensity and as varied in forms as that of encouraging a child to read. In fact, the variables have been designed in accordance with the cultural and socioeconomic feature of our developing country. In many provinces of Vietnam, that are more often rural than not, books are a luxury. Finance aside, it is also not common in the Vietnamese society for parents to personally select books for their children or to read for their children. For these reasons, an overly detailed variable in this aspect would only create unnecessary confusion for the instructors and students to answer adequately, especially given the cultural context previously explained.

## *1.4. Research Questions and Hypotheses*

This study aims to explore the relationship between demographic and socioeconomic factors with interest in reading books, as well as how personal habits regarding hobbies and book reading relate to self-reported interest for reading. Based on this, the following questions were formulated:

RQ1 What is the association between age, school grade, birth order, and reading interest?

RQ2 How does home literacy interact with the student's own reading interest?

RQ3 How do pastime activities influence reading interest? How does this factor interact with gender?

RQ4 How do books of different genres and the amount of time spent reading them relate to reading interest among students?

The hypotheses corresponding to these questions are as follows:

H1 Birth order and grade in school is negatively correlated with the propensity of taking an interest in reading.

H2 Students whose parents buy books for them and/or read stories out loud for them are more likely to take an interest in reading, compared to others.

H3 Preference for activities that require more introspection would be positively associated with reading interest regardless of gender.

H4    Time investment into books of both large themes—social sciences and natural sciences—has the
most positive effect on enhancing the student's likelihood to take an interest in reading.

## 2. Materials and Methods

### 2.1. Material

#### 2.1.1. Dataset

The dataset consists of 1676 observations, a subset of the data obtained through the survey
"Studying reading habits and preference of junior high school students in Vietnam" [70]. The study was
conducted by Vuong and Associates office, from December 2017 to January 2018. The survey concerns
all adolescent students (grade 6 through 9, which corresponds to age 11 to 15) enrolled in public junior
high schools in Ninh Binh Province, situated in the Northern part of Vietnam. The investigation was
conducted in the form of directly filled questionnaires. The majority of the questions are multiple-choice.

Participating students were instructed by their homeroom teachers, who have previously been
briefed by qualified personnel about the general significance of each notion in the questionnaire.
This ensures that the respondents' understanding of the term 'reading' is coherent.

On a provincial level, the original survey returned a dataset that could be considered a complete
sample of junior high students in Ninh Binh. If representativity is considered on a national level,
the data has been purposively sampled to represent junior high students from a typical province in
Northern Vietnam, which could arguably extend to all junior high students in Vietnam. The data has
also been obtained through convenience sampling: the research and survey team are based in the north
of Vietnam, while the most willing collaborators are from the province of Ninh Binh.

The 1676-observation subset employed for analysis is a result of the first phase of data-entering
process in the study and was not subjected to any selection.

#### 2.1.2. Variables

We first investigated the students' interest in the activity of reading through the direct question:
"Do you like to read?", coded as variable 'Readbook,' with two values of 'yes' and 'no.' Over 90% of
students answered "yes." This means that the majority of students self-report as being interested in
reading (see Table 1).

The variable 'Readbook' was analyzed as the main dependent variable in this study.

The analyses would also contain the following independent variables:

- "Sex": the gender of respondent, with two categories: "male" and "female";
- "Grade": the school grade of the respondent, with four categories: "gr6", "gr7", "gr8", "gr9",
  representing grades 6 to 9 respectively. This variable could be equated to the age of the respondents
  relative to each other;
- "RankinF": the birth order of the respondent in their family. For example, if the student is a third
  child, they would answer with "3";
- "Hobby": the respondent's favorite past time. This variable has 6 categories: reading ("a"),
  watching TV or listening to music ("b"), helping with chores ("c"), observing nature ("d"),
  socializing with friends ("e"), and others ("f").
- "Buybook": whether or not the respondent's parents buy books for them, with two answers:
  "yes" and "no";
- "Readstory": whether or not the respondent's parents read books for them, with two answers:
  "yes" and "no";
- "TimeSci": time per day spent reading natural sciences books (self-reported), with two categories:
  under 30 min ("less30") and 30 min or over ("g30");
- "TimeSoc": time per day spent reading social sciences and humanities books (self-reported),
  with two categories: under 30 min ("less30") and 30 min or over ("g30").

A complete list of variables and their explanation, as well as other details concerning the design of the study and the original questionnaire could be found in Vuong, Le, La, Vuong, Do, Vuong, Do, Hoang, Vu, Ho and Ho [70].

**Table 1.** Distribution table of some categorical variables.

| Code Name | Explanation | Items | Frequency | Proportion |
|---|---|---|---|---|
| Grade | Current grade | Grade 6 | 467 | 27.86% |
| | | Grade 7 | 443 | 26.43% |
| | | Grade 8 | 410 | 24.46% |
| | | Grade 9 | 356 | 21.24% |
| Sex | Biological gender | Male | 853 | 50.89% |
| | | Female | 823 | 49.11% |
| Hobby | Favorite pastime (self-reported) | Reading | 331 | 19.75% |
| | | Watching TV, listening to music | 790 | 47.14% |
| | | Helping with chores | 180 | 10.74% |
| | | Observing nature | 59 | 3.52% |
| | | Socializing | 111 | 6.62% |
| | | Others | 205 | 12.23% |
| Buybook | Whether respondents' parents buy a book for them | Yes | 1447 | 86.34% |
| | | No | 229 | 13.66% |
| Readstory | Whether respondents' parents read books for them | Yes | 424 | 25.30% |
| | | No | 1252 | 74.70% |
| TimeSci | Time per day spent on natural sciences books | Under 30 min | 846 | 50.48% |
| | | 30 min or over | 830 | 49.52% |
| TimeSoc | Time per day spent on social sciences and humanities books | Under 30 min | 1065 | 63.54% |
| | | 30 min or over | 611 | 36.46% |
| Readbook | Answer to question: "Do you like reading books?" | Yes | 1512 | 90.21% |
| | | No | 164 | 9.79% |

## 2.2. Methods

Raw data were entered in an MS Excel spreadsheet and converted into CSV file type. Data analysis was done in R. The baseline-category logit (BCL) model was employed to explore the relationship between pairs of variables on the dependent variable. Similar usage of BCL can be found in [71,72]. To estimate how changes in the values of independent variables impact the dependent variable, logistic regression was used to predict the probability of a category of dependent variable Y against different values of independent variables **x**. Estimate coefficients were calculated through multinomial logistic regressions and later used to calculate conditional probabilities.

The general equation of the logistic regression model is as follows:

$$\ln \frac{\pi_j(\mathbf{x})}{\pi_J(\mathbf{x})} = \alpha_j + \beta'_j \mathbf{x}, \ j = 1, \ldots, J - 1.$$

in which **x** is the independent variable; and $\pi_j(\mathbf{x}) = P(Y = j | \mathbf{x})$ are the corresponding. $\pi_j = P(Y_{ij} = 1)$ with $Y$ being the dependent variable.

The beta coefficient reflects the relationship between the corresponding independent variable **x** and the logit of dependent variable $Y$. When $\beta > 0$, larger values of **x** are associated with larger logits of $Y$ and vice versa. When $\beta < 0$, larger values of **x** are associated with smaller logits of $Y$. $\beta = 0$ is considered the null hypothesis, which states that there is no relationship between **x** and $Y$ in the population [73].

The probability of the values of the dependent variable is calculated as follows:

$$\pi_j(\mathbf{x}) = \frac{\exp\left(\alpha_j + \beta'_j \mathbf{x}\right)}{1 + \sum_{h-1}^{J-1} \exp\left(\alpha_j + \beta'_j \mathbf{x}\right)}$$

With $\sum_j \pi_j(\mathbf{x}) = 1$; $\alpha_J = 0$ and $\beta_j = 0$; in which $n$ is the number of observations in the sample, $j$ are the categorical values of an observation $i$, and $h$ is the number of rows in matrix $\mathbf{X}_i$.

In this research, the dependent variable in all models is "Readbook." The statistical significances of the models are assessed based on $z$-value and $p$-value ($p < 0.1$ was chosen as the threshold for statistical significance). All four models present in this paper employed the same regression method.

While it is technically possible to run a single regression of all independent variables present in this paper against the dependent variable "Readbook", we have made a conscious decision to split our analyses into four models. This allowed us to focus on the effect of specific pairs of independent variables in relation to each other as well as to the response variable, as fitted models offer more precision [74].

## 3. Results

### 3.1. Descriptive Statistics

The data shows that "RankinF" ranges from 1 to 7; the largest total number of children in a family is reported to be 8. This means that the highest birth rank reported by our respondents is 7. In other words, none of the respondents is the youngest child in their family. The number of male and female students is relatively similar. Students are also distributed rather evenly between the four school grades, with sixth graders taking the largest share in the sample (~28%).

Despite the fact that 90% of students reported taking an interest in the activity of reading, only 20% considered reading to be their favorite hobby. The modal category was "watching TV or listening to music." Females tend to prefer doing chores and observing nature more than males (see Table 6).

Regarding the type of books being read, books on natural sciences seemed to be favored: nearly half the number of respondents spend 30 or more on this type of books, whereas only one-third of students spend the same amount of time on social sciences and humanities books. The data also showed that sixth grade students spend the most time reading both kinds of books, compared to their seniors. In contrast, seventh grade students spend the least time on this activity (see Appendix A, Table A1).

Over 86% of respondents reported that their parents bought books for them; however, only 25% said their parents read books for them.

### 3.2. Regression Results

#### 3.2.1. RQ1—H1

*Does their birth order in the family affect a student's propensity to like reading books? What is the relationship between age (inferred from school grade), birth order, and reading interest?*

In this first logistic regression model, the dependent variable "Readbook" was analyzed against two independent variables, "RankinF" and "Grade". "RankinF" was treated as an ordinal variable. Results are displayed in Table 2.

Table 2 shows that all estimate coefficients are statistically significant ($p < 0.1$) and the null hypothesis is rejected. This means that there is a relationship between the birth order and grade of the respondents and their interest in reading.

**Table 2.** Estimate results of "Readbook" by "RankinF" and "Grade".

| | Intercept | "RankinF" | "Grade" | | |
| --- | --- | --- | --- | --- | --- |
| | | | "gr7" | "gr8" | "gr9" |
| | $\beta_0$ | $\beta_1$ | $\beta_2$ | $\beta_3$ | $\beta_4$ |
| Logit (yes\|no) | 3.543 *** [12.218] | −0.153 [−1.930] | −1.091 *** [−3.728] | −0.807 ** [−2.626] | −1.686 *** [−5.924] |

Significance codes: 0 '***' 0.001 '**' 0.01 '*' 0.05; z-value in [square brackets]; baseline category for: "Grade" = "gr6". Null deviance: 1043.89 on 1660 degrees of freedom; Residual deviance: 994.06 on 1656 degrees of freedom; AIC: 1004.1.

The relationship between the variables is presented in the following Equation (1):

$$\ln\left(\frac{\pi_{yes}}{\pi_{no}}\right) = 3.543 - 0.153 \times \text{RankinF} - 1.091 \times \text{gr7} - 0.807 \times \text{gr8} - 1.686 \times \text{gr9} \tag{1}$$

From Equation (1), the conditional probability of each value of "Readbook" can also be calculated. For example, the probability of "yes" in "Readbook" at "RankinF" = 1 and "Grade" = "gr7" is:

$$\pi_{yes} = \frac{e^{(3.543-0.153\times1-1.091\times1)}}{1 + e^{(3.543-0.153\times1-1.091\times1)}} = 0.909$$

This probability measure means that a seventh-grade student who is the oldest child in their family is 90.9% likely to take an interest in reading. Similarly, other probabilities were calculated and presented in Figure 2 (Table of detail probability is in Appendix B, Table A2).

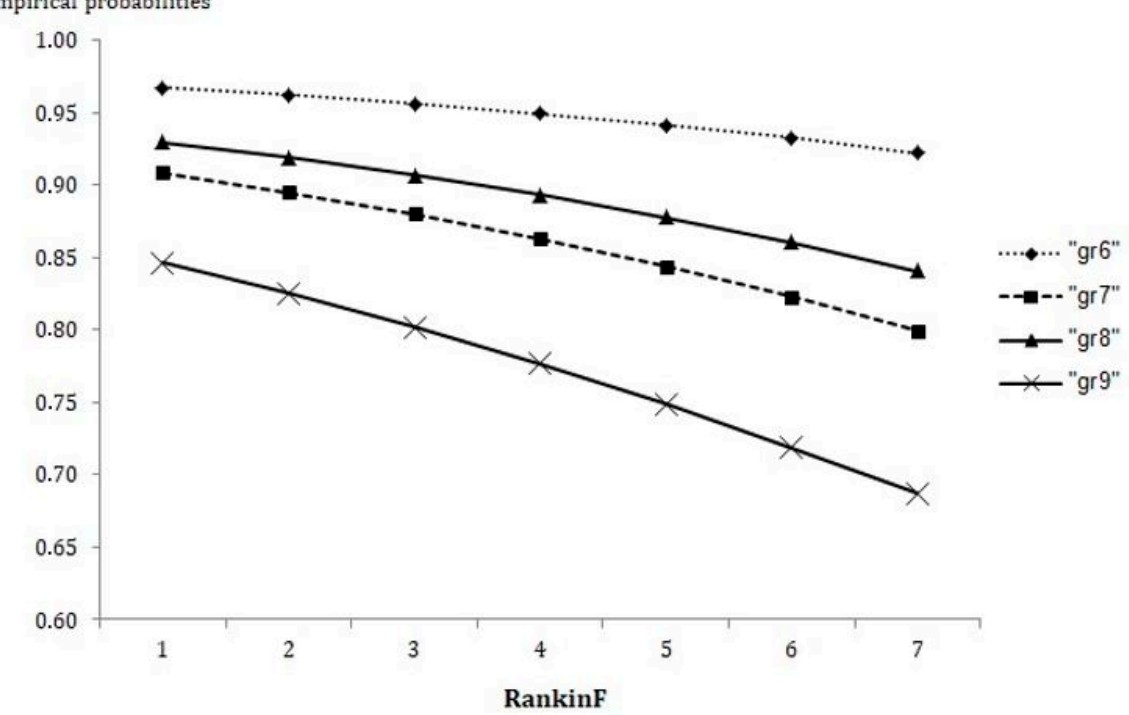

**Figure 2.** Probability of reading interest by school grade and birth order in the family.

The great majority of junior high students in Vietnam self-report as liking to read books. Figure 2 presents the probability of "Readbook" = "yes" across different categories of "Grade" and "RankinF". All lines are above 65%, meaning that students are likely to answer "yes" to the question "Do you like reading books?" regardless of their grade in school or their birth order in their family. This could be

interpreted as Vietnamese students either having a high propensity for intensive reading habits or simply having a tendency to self-report as liking to read.

It is easy to observe that "gr6" is the highest line, and "gr9" is the lowest. This implies that younger students are more likely to answer "yes" to the question, "Do you like to read?" (the highest being 92–97% for sixth graders) than older students (69–85% for ninth graders, the lowest). This aligns with the findings in the literature: reading practices and enjoyment decline with age. It appears that Vietnamese students are not exempt from the general global tendency.

It should also be remarked that the gap between highest and lowest "Readbook" = "yes" probabilities is the largest for "gr9" (15.9 percentage points) and the smallest for "gr6" (4.5 percentage points). In other words, the decline of reading interest is aggravated by the respondent's lower birth order in their family. This prompts for a further more in-depth view of the variable "RankinF". When moving from value 1 to value 7 of variable "RankinF", it could be easily observed that all lines "gr6", "gr7", "gr8" and "gr9" descend monotonically. This means that the more older siblings a student has, the less likely are they to self-report as liking to read. For example, a ninth-grade student who is a firstborn in the family would be 85% likely to report that they like reading, whereas a ninth-grade student who is the 7th child in the family would be only ~69% likely to report that they like to read.

A similar model examining dependent variable "Readbook" against "Grade" and "Hobby" points to the same tendency (see Appendix C, Table A6).

### 3.2.2. RQ2—H2

*Does the parents' act of buying books and reading books to their children affect the student's reading habits? In which direction and how strong is the relationship?*

To study the parents' role in cultivating for their children an interest in book reading, this model employed "Readbook" as the dependent variable and "Buybook" and "Readstory" as independent variables, with statistically significant estimate results ($p < 0.01$) as shown in Table 3. There is thus a relationship between having parents who buy and/or read books for oneself and taking an interest in reading books.

**Table 3.** Estimate coefficients of "Readbook" against "Buybook" and "Readstory".

|  | Intercept | "Buybook" | "Readstory" |
|---|---|---|---|
|  |  | "Yes" | "Yes" |
|  | $\beta_0$ | $\beta_1$ | $\beta_2$ |
| Logit (yes\|no) | 1.391 *** [8.146] | 0.841 *** [4.293] | 0.753 ** [3.159] |

Significance codes: 0 '***' 0.001 '**' 0.01 '*' 0.05; *z*-value in [square brackets]; baseline category for: "Buybook" = "no", "Readstory" = "no". Residual deviance: 0.0019 on 1 degrees of freedom. Log-likelihood: −9.6473 on 1 degrees of freedom.

The null hypothesis $\beta_1 = \beta_2 = \ldots = 0$ was rejected with $p-\text{value} = 1.115 \times 10^{-7} \approx 0$, which proved that the model is appropriate. The regression equation was presented as follows:

$$\ln\left(\frac{\pi_{yes}}{\pi_{no}}\right) = 1.391 + 0.841 \times \text{yesBuybook} + 0.753 \times \text{yesReadstory} \qquad (2)$$

Using this equation, probabilities of "Readbook" values against variables "Buybook" and "Readstory" were calculated (Table of detail probability is in Appendix B, Table A3). It could easily be seen that when parents actively buy and read books for their children, the likelihood of said children being interested in reading is 95.2%.

Figure 3 shows the role of parents in cultivating reading habits in students. A significant relationship can be observed as the dependent variable changes remarkably when moving from one value of the independent variables to another.

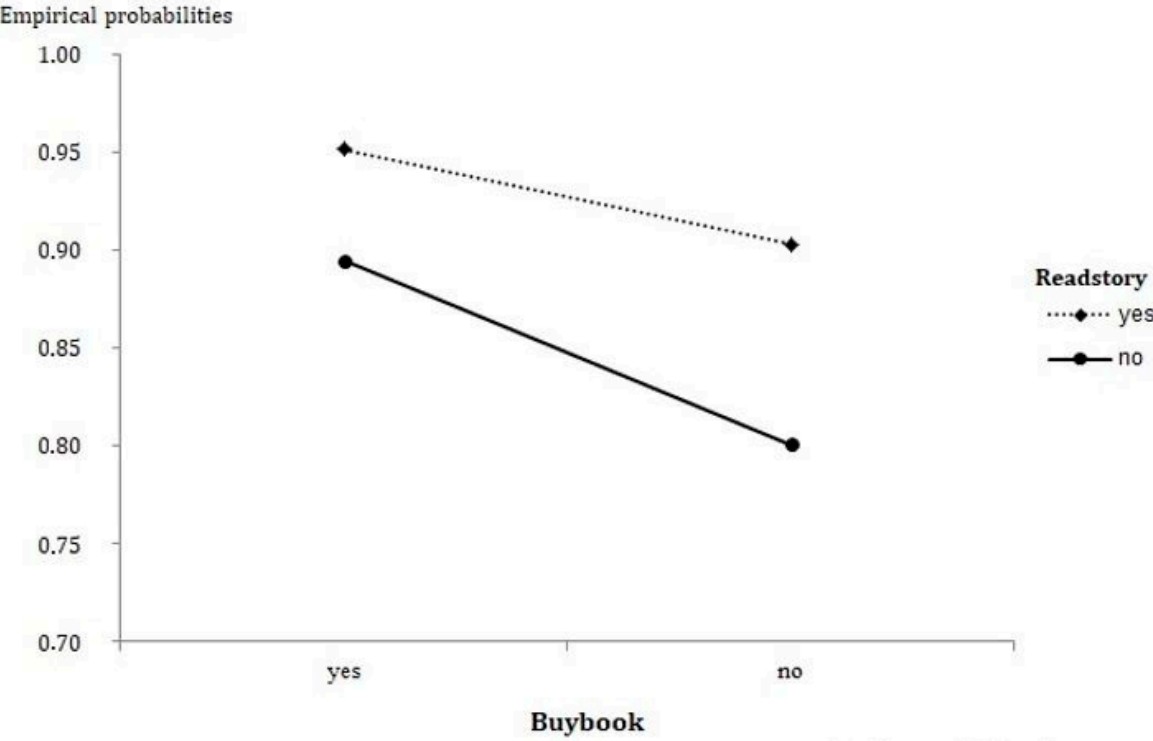

**Figure 3.** Probability of a student taking an interest in reading based on whether parents buy and read books for them.

The dotted line represents students whose parents have ever read books for them, whereas the other line represents those whose parents have never done so. The dotted line is well above, meaning that those who are often read to by their parents are much more likely to be fond of reading themselves. Similarly, parents buying books for their children also creates a positive influence in promoting reading interest. Namely, the probability of a student answer "yes" to "Readbook" at the point of "Buybook" = "yes" is ~90–95%, much higher than at "Buybook" = "no" (~80–90%). In addition, the effect of "Buybook" is stronger for students whose parents do not read for them: in fact, when moving from value "yes" to "no" of variable "Buybook", the "Readbook" = "yes" line only drops 4.9 percentage points whereas the "Readbook" = "no" line drops 9.4 percentage points. In other words, the combined effects of having parents read for them and having books bought for them by parents enhance the student's likelihood to form positive reading habits the most.

### 3.2.3. RQ3—H3

*How do gender and pastime activities influence reading interest?*

Gender and pastimes have a statistically significant influence on a student's interest in reading, as shown in the summary table of the logistic regression of "Readbook" by "Sex" and "Hobby" (Table 4).

As observed, all *p*-values are inferior or equal to 0.01; the null hypothesis is rejected and all coefficients are statistically significant.

Using the above estimate coefficients, the following regression equation is formulated:

$$\ln\left(\frac{\pi_{yes}}{\pi_{no}}\right) = \begin{aligned} &4.818 - 0.918 \times \text{Male} - 2.281 \times b\text{Hobby} - 1.613 \times c\text{Hobby} \\ &-1.948 \times d\text{Hobby} - 2.431 \times e\text{Hobby} - 2.383 \times f\text{Hobby} \end{aligned} \tag{3}$$

Equation (3) is then used to calculate conditional probabilities, displayed in Table A4 (Appendix B). Visualization of probabilities is shown in Figure 4.

**Table 4.** Estimate coefficients of "Sex" and "Hobby" on "Readbook". Note: a = reading; b = Watching TV/Music; c = Helping with chores; d = Observing nature; e = Socializing; f = Others.

| | Intercept | "Sex" | "Hobby" | | | | |
|---|---|---|---|---|---|---|---|
| | | "Male" | "b" | "c" | "d" | "e" | "f" |
| | $\beta_0$ | $\beta_1$ | $\beta_2$ | $\beta_3$ | $\beta_4$ | $\beta_5$ | $\beta_6$ |
| Logit (yes\|no) | 4.818 *** [9.362] | −0.918 *** [−4.879] | −2.281 *** [−4.419] | −1.613 *** [−2.719] | −1.948 ** [−2.823] | −2.431 ** [−4.214] | −2.383 *** [−4.384] |

Significance codes: 0 '***' 0.001 '**' 0.01 '*' 0.05; z-value in [square brackets]; baseline category for: "Sex" = "female"; "Hobby" = "a". Residual deviance: 10.934 on 5 degrees of freedom. Log-likelihood: −26.1477 on 5 degrees of freedom.

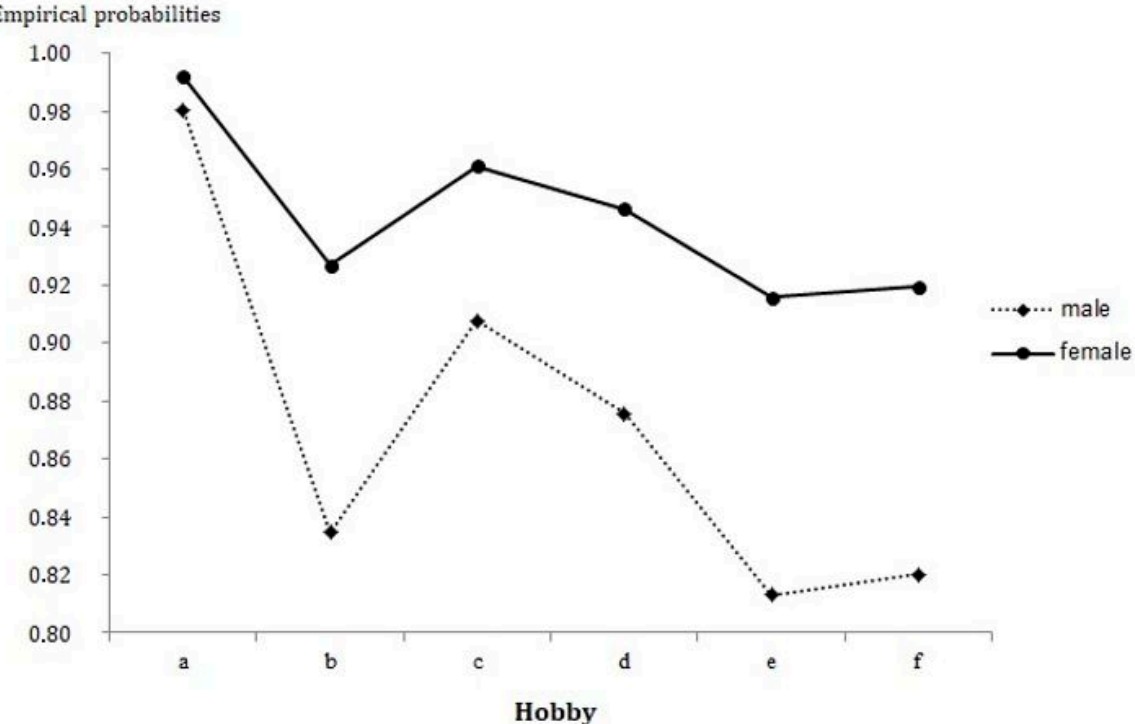

**Figure 4.** Probability of liking to read by gender and favorite hobby. Note: a = reading; b = Watching TV/Music; c = Helping with chores; d = Observing nature; e = Socializing; f = Others.

To have a better idea of how to interpret the probabilities, consider the following example: At "Hobby" = "e" and "Sex" = "male", the probability of "Readbook" taking the value "yes" is 0.813. This means that a male student whose favorite past time is to socialize with friends is 81.3% likely to report having an interest in reading. Similarly, one can see that a female student whose favorite hobby is reading is 99.2% likely to self-report as liking to read.

There is a remarkable difference between male and female students in terms of reading interest. In fact, male students are less likely to report that they like reading (82–89%). This is consistent with the extant literature on gender-based differences in students reading practices, such as Moffitt and Wartella [37] and Majid and Tan [33]. Rather than merely confirming the literature on female readers' predominance over males in reading avidity, these findings can also potentially bridge the gap between this and the smaller number of articles that had found out otherwise [30,39], by relating gender and favorite hobbies with reading interest in the same regression model.

On the one hand, regardless of gender, students who report reading as their favorite pastime are the most likely to reply "yes" to the question "Do you like to read?" (over 98%). In other words, considering reading as a favorite hobby is the strongest predictor to having great interest in reading. This is represented in Figure 3, along with several other findings. The second and third strongest

predictors for reading interest are "c"—helping with chores, and "d", being in nature. They appear to be the most introspective out of all the options given in the questionnaire and suggest a significant degree of introversion.

In contrast, the likelihood of reading interest is second lowest at "b"—watching TV or listening to music, which is also the modal category of a favorite hobby for both genders. This result might strongly relate to previous studies on the shifts in reading practices and interest in the context of multimedia development. Finally, the probability of the respondent taking an interest in reading drops to the lowest at "e"—socializing with friends. In light of this, the relationship between favorite hobby and reading interest might be related to one or more factors, just as the literature has suggested. But more importantly, we now have a semblance of the bridge we meant to build: a relationship between sex-based differential reading interest, and sex-based differential leisure interests.

The reported results highlight the gender-based differences in hobbies between female and male pupils, directly in relation to their reading practices. The effects of different pastimes on reading practices mentioned above are much starker in males compared to females, with the difference in terms of probability of "Readbook_yes" between the highest point ("a") and the lowest point ("e") being 16.7 percentage points for male students and 7.6 percentage points for female. It seems that the type of favorite hobbies is a stronger predictor for male reading habits than for female. This finding might have to do with the nature of the favorite pastimes themselves. Vuong, Le, La, Vuong, Do, Vuong, Do, Hoang, Vu, Ho and Ho [70] already reported that female and male differed in terms of hobby preferences. Table 5 presents the differences in more details.

**Table 5.** Distribution of male and female respondents by favorite pastimes.

| Favorite Pastimes | Male (%) | Female (%) |
|---|---|---|
| (a) Reading | 13.4 | 26.4 |
| (b) Watching TV/Listening to music | 52.4 | 41.7 |
| (c) Helping with chores | 8.8 | 12.8 |
| (d) Observing nature | 3 | 4 |
| (e) Socializing with friends | 6.4 | 6.8 |
| (f) Other | 15.9 | 8.4 |

More than half of the male pupils reported multimedia (TV or music) as their preferred pastime, while the second most common favorite hobby appears not to be listed among the options provided in the questionnaire and likely pertains to sports or video games. Female pupils, on the other hand, are much keener on reading and helping with chores. As such, our results show clear evidence of a gender-based difference in interest.

### 3.2.4. RQ4—H4

*Is there a correlation between the amount of time spent on reading social sciences and humanities books and on natural sciences books on one hand, and reading interest on the other?*

To test the relationship between duration of time spent on reading books and interest in book reading, a logistic regression model was applied to independent variables "TimeSci" and "TimeSoc", and dependent variable "Readbook". The results are shown in Table 6:

As $p < 0.001$, all coefficients are shown to be statistically significant. From this, Equation (4) was formulated as follows:

$$\ln\left(\frac{\pi_{yes}}{\pi_{no}}\right) = 3.566 - 1.049 \times \text{less30TimeSci} - 0.919 \times \text{less30TimeSoc} \tag{4}$$

Using this equation, specific conditional probabilities could be calculated. For example, a student who spends less than 30 min on natural sciences books and over 30 min on social sciences and humanities books is 92.5% likely to take an interest in the activity of reading.

$$\pi_{yes} = \frac{e^{(3.566-1.049\times1)}}{1 + e^{(3.566-1.049\times1)}} = 0.925$$

The probabilities were visualized in Figure 5 (Table of detail probability is in Appendix B, Table A5).

**Table 6.** Estimate coefficients of "Readbook" against "TimeSci" and "TimeSoc".

| | Intercept | "TimeSci" | "TimeSoc" |
|---|---|---|---|
| | | "Less30" | "Less30" |
| | $\beta_0$ | $\beta_1$ | $\beta_2$ |
| Logit (yes\|no) | 3.566 *** [15.580] | −1.049 *** [−5.268] | −0.919 *** [−3.956] |

Significance codes: 0 '***' 0.001 '**' 0.01 '*' 0.05; *z*-value in [square brackets]; baseline category for: "TimeSci" = "g30", "TimeSoc" = "g30". Residual deviance: 5.6254 on 1 degrees of freedom. Log-likelihood: −12.7051 on 1 degrees of freedom.

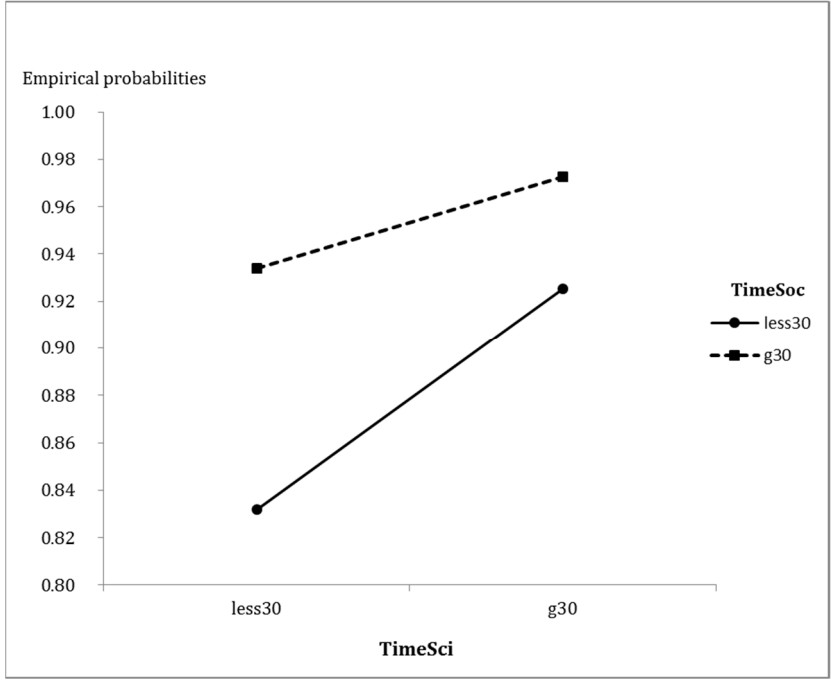

**Figure 5.** Probability of a student taking an interest in reading based on the amount of time they spend reading social sciences and humanities books and natural sciences books.

Figure 5 shows the propensity for self-reported preference for reading based on time spent on reading different types of books. It could easily be remarked that the combined effects of "TimeSoc" = "g30" and "TimeSci" = "g30" lead to the highest probability of liking to read. In other words, the more students spend time reading books, regardless of topics, the more likely they are to self-report as liking to read books.

Here, the result suggests that in order to develop reading habits of children, simply getting them to start reading would already provide a good entry point. More concretely, this means facilitating access to books, as well as promoting a culture of reading at school as well as at home. This also ties in with the findings in the previous section, according to which children whose parents buy and read books for them have a higher likelihood of being interested in reading practices.

More importantly, the different slopes of the two lines point out the incremental effect of spending more time to read, regardless of the type of books. In fact, if a student has already spent more than 30 min reading social sciences and humanities books, then whether they spend a lot of time (more than

30 min) or little time (less than 30 min) reading natural sciences books has little effect on the probability of them being fond of reading—only 4 percentage points (from 93 to 97%). On the contrary, if they spend little time reading social sciences and humanities books, then simply by spending more than 30 min reading natural sciences books would increase the likelihood of them liking the activity of reading by over 10 percentage points (from about 83% to nearly 93%). In other words, it does not matter whether students like to read literary works or popular science; as long as they spend more time reading books, even just one type of books, they would be more likely to report taking an interest in reading.

## 4. Discussion

### 4.1. Age and Birth Order

The reduced interest in reading among junior high school students has been documented before in a longitudinal study [30]. There could be, in fact, many reasons for pupils to lose their taste for reading. Given the dense curriculum in Vietnamese secondary education, it is often the case that older students have a heavier workload, thus less time for leisure reading. Besides, 9th-grade students see an especially severe drop in reading interest compared to younger students. This might be explained by the fact that 9th grade is the senior grade of junior high school in the Vietnamese educational system, and students are expected to take a senior high school entrance exam that is national and selective. This exam is widely considered to be even more important than the entrance exam for universities, therefore even more competitive. As such, 9th-grade pupils are obligated to spend a lot of time studying at school, at home, and in cram schools, which leaves them little time for leisure activities in general. They also often face harsh pressure from school, which might turn them away from any activities reminiscent of studying or activities that require more introspection, such as reading—a phenomenon that Mokhtari, Reichard and Gardner [26] has noted among college students. Finally, the decline in interest for reading could be a result of pupils' priorities changing with age: namely, older students are more pressured by the high school entrance exams and might be less inclined to spend their free time with books, which were intensively associated with schoolwork.

Regarding the association between the size of the family and the birth order of the child, our results appear to support the dilution model [49,50,52], with a specific focus on reading practices rather than merely child education in general. Our findings show evidence for the inverse relationship between pupil's reading interest and birth order, regardless of their school grade. A junior high-school-age adolescent is progressively less likely to take an interest in reading as they go down the ranks in terms of birth order. In other words, younger siblings in families with more children receive less attention from parents and thus are less likely to cultivate reading habits since childhood.

A feature worth noting is that the traditional household in Vietnam is often a three-generational extended family, consisting of the grandparents and at least one nuclear family unit of their offsprings. As such, the presence of extended family members should have an effect on the parental resource and family education, sometimes through delegation of responsibility of education [75,76]. Residence with the extended family could, in some case be a strategic choice to make use of parent aid [77]. The support provided by the delegation of caretaking tasks from parents to grandparents or uncles/aunts [76] could have in fact mitigated the dilution effect. However, one must take into account the deeply Confucian culture in the Vietnamese society [78–80], in which the first child, especially when said child is male, has a particular significance. This could cancel the positive effect of the extended caretaking resources of the three-generational family, leading back to unequal distribution of parental resource to the detriment of latter-born siblings.

The results presented in this section hint at the role of the family in cultivating reading habits among pupils. They also suggest a deeper look into the relationship between familial background in children's reading habits and, broadly speaking, education, all of which can very well be tested using the dataset prepared by Vuong, Le, La, Vuong, Do, Vuong, Do, Hoang, Vu, Ho and Ho [70]. Studies

have shown that the incremental effect of parent-child storybook reading diminishes as the child grows older, even in a short timespan such as from 2–3 years old to 4–5 years old [81]. As such, we went on to examine child reading interest in relation to family literacy. More specifically, we focused on parental provision of reading material as well as parent-child interaction in reading activities, in the section that follows.

## 4.2. Family Literacy

Previous findings have indeed emphasized on the significant role of cooperation between home and school in regards to developing reading abilities among children, more precisely the role of parents in the forming and encouraging of reading practices since childhood [42–44,81,82]. Where the vast majority of the extant literature focused on early childhood when it comes to parent-child reading experiences, this paper contributes results pertaining to junior high school-age adolescents. A point worth noting is that the likelihood of a student taking an interest in reading is the highest when their parents both buy books for them and read for them. This suggests that merely providing the material for children—namely, by buying them books—would not be as effective in creating a culture of reading in the family, compared to when parents actively guide their children into the world of reading. Despite this, the practice of reading for children is not widespread in Vietnam. In fact, while 86.3% of respondents have books bought for them by their parents, only 25.3% reported that their parents read books for them.

This is especially meaningful because children who received encouragement to read and access to book are more likely to have better educational records due to the development of skills and cognitive ability [22,83,84]. Storybook reading by parents is positively related to the development of skills related to language and reading in children [82]. Also, storybook reading by parents has been shown to be linked with reading motivation, especially in the case of struggling children [42]. In other words, having the family cooperate with the school would be optimal in initiating reading practices, forming reading habits, improving reading competency, and ultimately enhance the cognitive and academic ability of pupils.

## 4.3. Gender and Pastime

It is thus very often suggested that both biology and socialization—whether one factor dominates the other or not—come into play in determining sex-based differences in behavior and choices. More importantly, what matters to us is that there are such things as gendered, differential behaviors, namely in terms of leisure activity preferences, as shown in our results. In congruence with past findings, our data show a difference in male versus female hobbies, which is linked to male versus female reading interest. Our line of distinction (rather than Things versus People) is drawn between activities of high sensory stimulation (socializing with friends, watching TV/listening to music) and low sensory stimulation (reading, observing nature, helping with chores). Other than the fact that both boys and girls reported watching TV/listening to music as their favorite hobby, boys tend to adhere to the first group, whereas girls are more inclined to the second. It is also worth noting that girls are significantly more likely to report reading as their favorite leisure activity. More in-depth research would be necessary to establish any significant relationship between gender and types of leisure interest, as well as how it is linked to reading interest and practice.

A more thorough discussion on natural science versus humanity and social sciences in terms of book content and time spent on books follows in the next section.

## 4.4. Time Spent Reading

Regarding the genre of books, our model does not compare the influence of time spent on natural sciences books versus that on social sciences and humanities books separately; neither does it explore the role of time spent on reading books of either type on reading practices in regards to other factors such as age or gender. Rather, we compare the incremental values of spending more time on another type of

book when considerable time has been spent on one type of book. In other words, the emphasis lies on the variety of books being read. Even when respondents are avid readers of one type of books, their reading interest is still lower than respondents who spend a substantial amount of time (over 30 min) on both types of books. In other words, combining reading science and literary books strengthens the positive relationship that extended reading time had with reading interest. This suggests that in promoting reading, one must take into consideration not only the quantity but also diversification of reading materials. Previous findings suggested that reading fiction books had improved readers' empathy and social cognition [85,86]. Topping, Samuels and Paul [38] found that students who read fiction books are at a higher reading level compared to those who read non-fiction; they also boast higher reading quality. Non-fiction books were found to be more challenging and negatively correlated with reading achievement gains.

In addition, this also suggests that diversity of book genre might reduce boredom and enhance the reading experience, which could in turn be a solution to decline in reading enjoyment. Moreover, diversification of book genres could fit in more themes/subjects in terms of content, thereby tailor to the interest of different personalities. In the context of Vietnam, as the world of arts is being neglected [87], a strong emphasis on fiction and art books would generate interest among younger populations, teach children how to understand art, the craft of art-making, and foster a strong appreciation for the beauty of the world. Meanwhile, science, philosophy and technology books would introduce the younger populations to science, help them understand how science works, and the importance of scientific contributions to the world, and prepare for the future scientists.

### 4.5. Strategies for Sustainable Reading Culture

In order to promote sustainable reading habits in junior high school students, the collaboration among government, school, and family is important. Firstly, the government has a leadership role in promoting education for sustainable development. For instance, the funding to establish more public and in-school libraries and improve the quality of library services are necessary actions for the goals of sustainable education. Our results have shown that passive encouragement to books, represented by the provision of books to students by their own parents at home, already enhanced the likelihood of a student being interested in reading. Not all families could afford a variation of books, or are aware of the benefits of book reading, be it in terms of academic success, cognitive development or even simply personal enjoyment. While public facilities would not and should not aim to replace family literacy, well-equipped and accessible libraries would at least smooth out the differences between urban, more privileged students, and students in less developed areas.

Secondly, a reading culture fostered by the school helps to promote equitable education, which is one of the goals of sustainable development specified by UNESCO. Thus, schools have a vital role in stimulating students' interest to learn and lifelong learning skills. These results also suggest a review of the school curriculum and pedagogy. Schools should, in fact, not only focus on promoting interest in studying and cultivate a taste for knowledge seeking among students, but also seek to sustain such interest. As such, it is not the quantity of knowledge or school subjects that should take center stage, but the student's enjoyment. In essence, studying should not be a factor that distance students from activities that could have been recreational but has become associated with school and thus lost appeal—namely, in this case, the activity of leisure reading.

It should also be noted that junior high school pupils in Vietnam intensively associate books with school. Based on the dataset published in Vuong, Le, La, Vuong, Do, Vuong, Do, Hoang, Vu, Ho and Ho [70], a large number of pupils still cited school textbooks (such as 7th Grade Literature, for example) when asked about their favorite leisure books, with an explicit requirement to exclude school textbooks. This suggests that Vietnamese pupils attribute reading to schoolwork more than to recreation, and also that their repertoire is rather limited. Hence, schools should develop and integrate different activities to stimulate students' extrinsic and intrinsic motivation for reading. Examples of measures include independent reading time in class; extensive reading as a part of assigned learning

activities and homework; organizing reading clubs and activities related to reading such as debating or writing competitions, exhibitions that encourage teachers and students to bring reading materials and draw book illustrations for display. Moreover, training programs for teachers and librarians to engage in reader development are also needed. In order to implement these changes successfully, the involvement of the government through regulations, laws, and policies is also necessary.

Finally, family and a scholarly culture at home have been proved to be important for fostering children's reading habits. Parents are the role models, motivators, and facilitators for their children. Joining children in reading activities, encouraging them to develop positive attitudes to print language, and maximizing their access to books from an early age are important factors that will promote lifelong learning behaviors. Policy-makers should consider the fact that reading promotion is not only about the student but also their familial environment, as evidenced by reports such as Clark and Foster [28]. Moreover, parents should not only be encouraged to read for their children but read with their children. Research studies have suggested that it is not merely the existence of the storybook reading experience, but the parent-child interaction during those experiences that determine the effectivity of shared reading time on promoting childhood literacy [47,82,88]. Yet, numerous studies such as have pointed out that parents do not intuitively seek dialogues and interaction when reading to their child; there is reading styles would change drastically if a form of instruction was given [89,90]. As such, it might be the case that campaigns for promotion of reading should not only focus on just the child reader but also include assistance or instruction to parents in regards to how they should carry out the experience of storybook reading.

As complex as the topic is, the findings raised in this paper and the corresponding policy implications have hardly touched on all the issues related to reading and elementary education in Vietnam, which future research could address. In regards to reading enjoyment and interest among junior high-age pupils, future studies may benefit from drawing on the extant research on different reading instruction techniques. For example, the majority of the literature on dialogic reading techniques, first developed by Whitehurst, et al. [91], has so far only focused on preschool-age children and their reading capacity development. Given its positive effect as supported by many studies throughout the years and across the globe [81], it would be of interest to attempt these techniques in the junior high-school age range. Regarding methodology, as has also been shown in our study, elements such as the subject's time spent on reading or interesting reading could prove difficult to measure without relying on self-report, which would then raise the question of the validity of assumptions. In this case, different statistical approaches such as Bayesian statistics could be used for data analysis [92], to provide results in which difficulties that are exclusively related the frequentist approach—such as the question of assumption, as stated above—would be mitigated or eliminated completely.

**Author Contributions:** Conceptualization, T.T. and Q.-H.V.; methodology, T.T. and Q.-H.V.; formal analysis, T.T., T.-T.N., T.-T.-H.L., A.-G.P., M.-H.N., M.-T.H., T.-H.V., T.-T.V.; data curation, T.-T.N., T.-T.-H.L., A.-G.P., M.-T.H., M.-H.N., T.-H.V., H.-M.V.; writing—original draft preparation, T.-T.V. and M.-T.H.; writing—review and editing, T.-T.V., T.-H.V., M.-T.H., Q.-H.V., M.-H.N., P.-H.H., and H.-M.V.; visualization, T.-T.N., T.-T.-H.L., A.-G.P., M.-H.N., T.-H.V.; supervision, Q.-H.V. and T.T.; project administration, T.T. and Q.-H.V.

**Funding:** This research received no external funding.

**Acknowledgments:** We would like to send our gratitude to research staff of Vuong and Associates (Hanoi, Vietnam) for assisting in collecting data, especially Do Thu Hang, and Dam Thu Ha. Our most sincere thanks also go on to personnel of junior high schools and provincial departments that provided support during the survey.

**Conflicts of Interest:** The authors declare no conflict of interest.

## Appendix A. Cross-Tabulation of "TimeSoc", "TimeSci" and "Grade"

**Table A1.** Cross-tabulation of "TimeSoc", "TimeSci" and "Grade".

| TimeSoc | TimeSci | Grade | | | |
|---|---|---|---|---|---|
| | | 6 | 7 | 8 | 9 |
| **g30** | g30 | 138 | 108 | 123 | 68 |
| | less30 | 49 | 39 | 52 | 34 |
| **less30** | g30 | 133 | 73 | 97 | 90 |
| | less30 | 147 | 223 | 138 | 164 |

## Appendix B. Tables of Detail Probabilities for Each Research Question and Hypothesis

**Table A2.** Probabilities of "Readbook" against "Grade" and "RankinF" (RQ1—H1).

| "Readbook" | "Yes" | | | | | | | "No" | | | | | | |
|---|---|---|---|---|---|---|---|---|---|---|---|---|---|---|
| "Grade"\| "RankinF" | 1 | 2 | 3 | 4 | 5 | 6 | 7 | 1 | 2 | 3 | 4 | 5 | 6 | 7 |
| "gr6" | 0.967 | 0.962 | 0.956 | 0.949 | 0.941 | 0.932 | 0.922 | 0.033 | 0.038 | 0.044 | 0.051 | 0.059 | 0.068 | 0.078 |
| "gr7" | 0.909 | 0.895 | 0.880 | 0.863 | 0.844 | 0.823 | 0.799 | 0.091 | 0.105 | 0.120 | 0.137 | 0.156 | 0.177 | 0.201 |
| "gr8" | 0.930 | 0.919 | 0.907 | 0.893 | 0.878 | 0.860 | 0.841 | 0.070 | 0.081 | 0.093 | 0.107 | 0.122 | 0.140 | 0.159 |
| "gr9" | 0.846 | 0.825 | 0.802 | 0.776 | 0.749 | 0.719 | 0.687 | 0.154 | 0.175 | 0.198 | 0.224 | 0.251 | 0.281 | 0.313 |

**Table A3.** Probabilities of "Readbook" against "Buybook" and "Readstory" (RQ2—H2).

| "Readbook" | "Yes" | | "No" | |
|---|---|---|---|---|
| "Buybook"\|"Readstory" | "Yes" | "No" | "Yes" | "No" |
| "yes" | 0.952 | 0.903 | 0.048 | 0.097 |
| "no" | 0.895 | 0.801 | 0.105 | 0.199 |

**Table A4.** Probabilities of "Readbook" against "Sex" and "Hobby". Note: a = reading; b = Watching TV/Music; c = Helping with chores; d = Observing nature; e = Socializing; f = Others. (RQ3—H3).

| "Readbook" | "Yes" | | | | | | "No" | | | | | |
|---|---|---|---|---|---|---|---|---|---|---|---|---|
| "Sex"\|"Hobby" | "a" | "b" | "c" | "d" | "e" | "f" | "a" | "b" | "c" | "d" | "e" | "f" |
| "male" | 0.980 | 0.835 | 0.908 | 0.876 | 0.813 | 0.820 | 0.020 | 0.165 | 0.092 | 0.124 | 0.187 | 0.180 |
| "female" | 0.992 | 0.927 | 0.961 | 0.946 | 0.916 | 0.919 | 0.008 | 0.073 | 0.039 | 0.054 | 0.084 | 0.081 |

**Table A5.** Probabilities of "Readbook" against "TimeSci" and "TimeSoc" (RQ4—H4).

| "Readbook" | "Yes" | | "No" | |
|---|---|---|---|---|
| "TimeSci"\|"TimeSoc" | "Less30" | "g30" | "Less30" | "g30" |
| "less30" | 0.832 | 0.925 | 0.168 | 0.075 |
| "g30" | 0.934 | 0.973 | 0.066 | 0.027 |

## Appendix C. Estimate Results of "Readbook" by "Grade" and "Hobby"

**Table A6.** Estimate results of "Readbook" by "Grade" and "Hobby".

| | Intercept | "Hobby" | | | | | |
|---|---|---|---|---|---|---|---|
| | | "Grade" | "b" | "c" | "d" | "e" | "f" |
| | $\beta_0$ | $\beta_1$ | $\beta_2$ | $\beta_3$ | $\beta_4$ | $\beta_5$ | $\beta_6$ |
| Logit (yes\|no) | 7.109 *** [9.116] | −0.376 *** [−4.692] | −2.306 *** [−4.471] | −1.457 * [−2.453] | −1.919 ** [−2.783] | −2.445 *** [−4.242] | −2.373 *** [−4.362] |

Significance codes: 0 '***' 0.001 '**' 0.01 '*' 0.05; *z*-value in [square brackets]; baseline category for: "Grade" = "6"; "Hobby" = "a". Null deviance: 1073.77 on 1675 degrees of freedom; Residual deviance: 992.26 on 1669 degrees of freedom; AIC: 1006.3.

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
