# Peer review of "The Relationship between Birth Order, Sex, Home Scholarly Culture and Youths’ Reading Practices in Promoting Lifelong Learning for Sustainable Development in Vietnam"

_sustainability, doi:10.3390/su11164389_

Round 1

Reviewer 1 Report

Identifying a gap in research on reading habits, this paper is generally well-structured with clear questions and sound methodology. It uses a quantitative approach to elaborate upon socially relevant points, as advocated by Bourdieu (among others). The research effectively addresses the role of families in developing intellectual pursuits, it notes gender-based equity issues, and it touches upon the important point of encouraging diversity of reading materials. Beyond these generally positive attributes, several points can be addressed.

While the empirical approach seems sound, it is beyond my expertise to fully evaluate it; I will limit my comments to conceptual framework and assumptions.

The connection to sustainable development at times seems like an afterthought, bordering on gratuitous. It is referred to briefly in the discussion section. This makes the title appear misleading, so perhaps remove that element from the title (which is somewhat verbose). The mentioning of sustainable development in the discussion doesn't really add anything to the paper, and perhaps distracts from its other points, so that paragraph could be removed as well, and more could be made of other points noted in the discussion, such as social class and gender.

The literature review notes that, "Evidence relating to reading habits to age had been documented numerously in the past." However, the cited references to the past are primarily from the American experience, and "past" seems to go no further than 1980. So some clearer bookends could be applied to the literature review, in the interest of scope and with care to avoid over-generalization.

Books and reading are partially clarified by noting some differences between science and humanities. The paper would be more convincing if other points were addressed. Does the data refer to reading full books, or skimming? Are picture books included? Novels? Language is not clearly addressed, with the assumption being (perhaps unintended) that this research is only about English, or, more problematically, that what applies to English applies to all languages.

While statistics and the assumptions of the development discourse may take the above for granted, reading for pleasure or to spur imagination and reading for economic development are not the same. The research could therefore address the limitations of utilitarian reading for "knowledge and skills." Beyond that, to presume that reading serves economic development from early childhood seems heavy-handed, and begs the question of why not just put grade school children directly into vocational training. So, overall, the paper seems to be mixing the classical and utilitarian attitudes toward reading. This conceptual vagueness is heightened by considering textbooks, which are generally studied but not "read," in the same category as novels. So, a clearer distinction could be made between the goals and varieties of "reading," beyond just exposure to organized words.

Though Singapore and Malaysia are noted as presumably Asian English speaking societies (though this is not explicitly stated), most of the literature reviewed emanates from the American experience. Some attention to other cultures and societies, and languages, could improve the breadth and tighten up the conceptual framework. Even if the research is limited to English (it's not clear if the study is focusing on English only), India and Nigeria leap to mind as populous developing countries with legacies of English. Beyond focusing on nation states, there seems to be a lapse in attention to orality based cultures. Orienting the research mainly toward print literacy in a way that seems to exclude oral cultures needs to be at least explained. The taken for granted position (especially in development studies) is that oral cultures are "illiterate," implying some sort of defect. Even if the focus will remain on development and urban based industrialized societies, this could be made more explicit, with some attention to how development favors print-based literacy by undermining other literacies. So, a few sentences dealing with the pros and cons of print literacy and orality could be added in the interest of providing scope and balance.

It is not a central concern, but is somewhat troubling to equate readers with pupils. Better focus could be applied to the object of this research, i.e. if it is about young people attending school in societies that favor industrialization, then say so. The congenital relationship between schooling and industrialization has been ongoing since the 19th century and spread with colonialism, often at the expense of indigenous knowledges and livelihoods. There is a fairly thorough critical literature on development that ought to at least be noted if only to dismiss it as irrelevant to this study. As it stands now, omitting these points can amount to an unacknowledged set of assumptions that seems to favor development at the expense of other ways of knowing and living. This can be easily missed by the thoroughly development and heavily industrialized societies, but in the Global South (i.e. developing world), this cannot, and should not, be presumed.

For example, though beyond the scope of the present paper to address this in detail, it could be briefly noted that print-based literacy emanates from and favors urban, industrialized, technological over rural and agrarian societies, and that various forms of wisdom, such as how to live with nature, are undermined by print literacy. C. A. Bowers has written consistently and cogently on this topic.

At least a paragraph or two acknowledging these ideas will go a long way toward making the literature review more rigorous, and in noting the limits of the study, which as it stands now is weakened by a reliance on unexamined assumptions.

Beyond the above points on conceptual framework, details need attention. To name a few: Accuracy: The Ray "Bradley" quote is actually from Ray Bradbury, the American sci-fi novelist; Grammar: 70 million Vietnamese people own mobile phone [add indefinite article or make plural]; Typos and/or spelling: Journal of "Educaitonal" Media & Library Sciences [correct typo]. A thorough proof read and fact check will likely catch, and clean up, any other stylistic oversights.

Author Response

Dear reviewer 1,

We would like to sincerely thank you for your insightful review. Following your suggestions, we have done major revisions on our manuscript. You will find the modifications highlighted in yellow, and additional content highlighted in green. We will address each of the points you have made below, along with our response.

There was an apparent lack of connection between the body of our manuscript and the scope of the journal. We have thoroughly revised our manuscript and body of cited literature in order to address this issue. The discussion section has also been modified to more accurately articulate our ideas and relate the results of our analyses to the subject of sustainability.

The literature review has been updated, notably with works that did not subscribe to the model of development, as well as a brief literature concerning literacy. We would like to thank you for the many suggestions regarding the conceptual framework, in particular the nuanced view on different cultures and print-based literacy.

We have added a section focused on conceptualization, in which the notion of reading along with related theories and concept have been detailed. The aspect of language – which, in the case of this study, is the native language of the students, Vietnamese – has also been addressed, both in the literature and in the section on conceptual framework. Other methodology-related details such as the subject of the research have also been clarified.

Finally, we have taken a thorough revision of the paper in order to clean up grammatical and factual errors.

We believe we have addressed all of your concerns, and hope that you find our response and the revision satisfactory. Once again we would like to thank you for your in-depth remarks and detailed suggestions. It is our honor to have the opportunity to work with you to improve our manuscript.

With our highest respects,

The authors

Reviewer 2 Report

Please see the file attached.

Author Response

Dear reviewer 2,

We would like to sincerely thank you for your insightful review. Following your suggestions, we have done major revisions. In our manuscript, you will find the modifications highlighted in yellow, and additional content highlighted in green. We will address each of the points you have made, along with our modification, below.

It has been pointed out that our article was lacking in connection with the scope of the journal. We have thoroughly revised our manuscript and body of cited literature in order to address this issue. The discussion section has also been modified to better articulate our ideas as well as the connection between the results of our analyses and the subject of sustainability.

We have added a section dedicated to the conceptual framework of the study. Namely, we have explicitly defined the notion ‘reading’ as understood in the article, as well as related terms and variables that previously appeared superfluous. A formal delimitation of the subject of the research has also been added.

Certain variables have been put into question, regarding their capacity to measure what they were meant to indicate and the soundness of the data. We thank you for raising the issue and have detailed the justification for the study’s design in the section on concepts.

Passages concerning the methodology and technical details of the study have also been updated with descriptions and clarification that would better favor a general readership. The hypotheses and method of data collection have been explicitly stated. We thank you for the suggestion.

We believe that, throughout our revision, we have taken into account all of your concerns. We hope that you find our response and the revision satisfactory. Once again we would like to thank you for your in-depth remarks and detailed suggestions. It is our honor to have the opportunity to work with you to improve our manuscript.

With our highest respects,

The authors

Round 2

Reviewer 2 Report

Dear Authors,

Please find my suggestions in the attached file.

Author Response

Dear reviewer 2,

Once again we would like to express our most sincere thanks for your contribution in helping us improve our submission. We have done major revisions following your very detailed input. The changes brought to the manuscript have been highlighted in yellow, whereas newly added passages were highlighted in green. In this letter, we will address each of the points you have made individually.

Page 2, the illustration seems out of genre for a scientific journal.

We thank you for the feedback and apologize for the lack of justification for the illustration. We shall now address this addition which we consider substantial. First, with the use of a so-called “impossible object” figure as the basis of the illustration, we hoped to pay homage to renowned artist M. C. Escher whose works are known for, among other remarkable values, mathematically intriguing perspectives.

Second, the illustration has been created exclusively for the article; its theme and visuals have thus been conceived to purposefully reflect the theme and insights of our research. Even though this is a scientific article, we truly believe in the philosophical relations between aesthetics and knowledge. Through the use of an illustration, we hope to visually reflect the value of books and book reading, aesthetical and beyond.

Third, we maintain that imagination is a crucial element of doing science. This illustration, to us, represents this quality: the ability to turn childlike curiosity and speculations into scientific investigations that yield meaningful results. We hope we have made it clear that the time and efforts we put into this illustration came across as thoughtful.

Page 3, is seems unfortunate to rely on „unofficial” statistics in a scientific article. Also, it is not clear what the significance of the statistics being „unofficial” is. Either they are reliable or not.

We have clarified the sources of the statistics in the main text. The figures came from official government statistics, reported by the Ministry of Education and Training and Vietnam Publishers Association. We referred to them as “unofficial” only because we had to cite a secondary source due to a lack of formal documentation from the government.

Page 3, claims about the effectiveness of the Vietnamese education system (in spite of the indications that pupils seemingly do not read 'enough') are a little messy. The fact that the country has winners at the Mathematical Olympiad only proves that brilliant students exist in Vietnam, but it is not necessarily a good measure of the educational system as a whole. Conversely, if we do believe that this excellent result is indicative of the whole educational system, then it weakens the argumentation of the authors that reading is so important for cognitive development – after all, if pupils not reading 'enough' can win the Mathematical Olympiad, why should they read any more? This argumentation should be addressed.

We have removed the part on Vietnamese students at the Mathematical Olympiad. The argumentation is now focused on the contrast between the high score in Science and Mathematics and the low score in Reading, which was based on the Programme for International Student Assessment (PISA) test.

Page 3, Section 1.1 last paragraph: „What is the contribution of reading habits to Vietnamese students’ ability? To the best of our knowledge, no existing study has addressed this question. Moreover, reading habits and high school students are also neglected by researchers in the country.” – It is interesting to pose the first question that I quoted – after all, the present article does not answer it either. Also, I generally doubt that any country „neglects” its high school students (en bloc) in educational research, but I do see that I may be wrong there, and the authors can stick to their claim if they feel it is right. Still, maybe the claim can be refined to say that high schoolers' reading habits are not well researched.

We have edited the sentence to: “According to a recent study, reading habits and high school students are not well researched by researchers in Vietnam: only 19 out of 174 articles of Vietnamese researchers that were indexed in Web of Sciences are about general education [10].”

Page 3, Section 1.2 begins with a newly added, very fine theoretical underpinning of reading. It goes relatively deeply into cognitive psychology, which is fine, but does not relate strongly to the rest of the article. (Although I am not exactly saying that it should be changed.)

Thank you very much for your input. We believe the section would be helpful for our later definition of the term “reading”, which we will address in further details in the following bullet points.

Page 4 and onwards, I still find it important to stress that a definition of what constituted „reading” in the empirical studies should be added. Was it about book reading for leisure, was it about newspapers or any other material as well? Clarification would be welcome.

The notion of “reading” and its definition within the framework of our study has been addressed in Section 1.3. We have added another sentence for clarity. Please find the passage in question below. Please refer to Section 1.3. Notions and concepts, page 7, for further details.

“Respondents are allowed to apply their intuitive understanding of the term reading, with only minimal instruction in order to assure a global coherence between filled records. The activity could thus be understood as meaningful exposure to texts that are not necessarily intensive. This means that leisure reading is included and there is little limit on the reading material and subject. […] This open understanding of reading also means that any reading medium, i.e. paper-based, digital, etc. could be included.”

Page 4, the empirical studies reviewed should perhaps be described in more detail. Especially e.g. source 28 referring to a research with „panel data”. Who were followed? For how many years? Which country? What kind of „reading” was studied?

We thank you for the in-depth suggestions, following which we have revised our literature and enriched our literature review. We have opted to still leave out certain details that concern too narrowly the methodology of individual articles, eg. the type of “reading” studied in source 28 (now 29), in which the measurement of the student’s reading activities and their enjoyment thereof employed Teale-Lewis Reading Attitude Scales and the Reading Behavior Profile. This is a conscious choice in order not to render the passage superfluous and dilute attention on the main thread of argument. The reader is free to refer to the original sources for more details.

Page 4 bottom, the sentence „the motivations for different types of books – namely between information/expository books and narration/narrative books – are different” is not worth much if the difference is not explained.

Thank you for the remark. We believe this is an issue of language, as the phrasing of the sentence has diverted attention from the main argument we were presenting. We have therefore reviewed the passage, as follows: “In addition, the motivations for reading and by extension the level of enjoyment derived from reading may vary on an individual basis in relation to personal preferences regarding type of books [32,33].”

Page 5, the paragraph beginning „Demographic and socioeconomic” needs refinement, more detailed explanation of sources. What age groups are studied here? What does „third-grade” mean in the given cited source (e.g. elementary school, secondary school)?

We appreciate your suggestions and hope that the changes we have made to this paragraph are satisfactory:

Gender has also been examined in relation to reading patterns and practices. Research studies in both Western and Eastern societies have shown that males and females are often reported to differ significantly in reading enjoyment, motivation, and reading material preferences [27,30,32,36,37]. Most of these results show that girls read more and are more interested in reading than boys; however, findings based on data from 12 to 15 years old Australian students suggested otherwise [38]. In terms of race, Asians were slightly better at reading reports while White found reading newspapers and novels easier [39]. A recent research article showed that third grade at elementary schools female students demonstrated similar reading ability as their male counterparts, but they valued reading more [40].

Page 5, „Clark and Foster [26] showed that pupils who are eligible for free school meals, implying a background of lower socio- economic status, are less enthusiastic readers; their parents also read less or own fewer books.” – It might be interesting to know whether Clark and Foster offered a hypothesis as to what the 'true cause' in that relationship might be. Is it book ownership/access (whereby if a 'rich' family owned no books, their kids would also not be avid readers), or is it rather parental practice (whereby if in a rich family with many books even, parents did not read, it would lead to the same result)?

Clark and Foster did not offer a hypothesis as to what the ‘true cause’ in the relationship because this is a report from National Literacy Trust that reported the results of a survey. Even though the authors did not offer any hypothesis, they recommended schools to create of culture that foster enthusiastic readers, encourages boy to read, and support parents in reading to children at home.

Page 5, „Compton-Lilly [47] had, based on longitudinal, qualitative data, gained insights into the role of familial and socioeconomic context on forming children’s discourse and literacy.” – Such a sentence does not serve the article well if the actual insights are not described. If they are not important for this study, then the source should not be mentioned.

We have modified the passage accordingly.

Page 5, the literature on birth order/family size is now part of the literature review, which I had suggested in my first review. I welcome this change. However, it is a sudden change in the text – from empirical literature on reading to suddenly childrearing issues in general. A sentence that justifies the inclusion of this section, something that integrates it to the overall argumentation of the literature review, should be added.

We have added a transition that would tie the passage into the section. Thank you for the input.

Page 6, again a more detailed description of a source is needed, „Ennemoser and Schneider [45] based on longitudinal data” – which country, what age group are we talking about? The text mentions a „three-year gap” which is not easy to understand; should be clarified.

We have clarified the source as follow: “Ennemoser and Schneider [45] based on the data from 332 German children with measures collected from 1998 to 2001, suggested that TV viewing had at least a medium-term effect on reading ability (three-year gap), especially at an early age.”

Page 6, „(it would have been a bleary day if such a binary is all that is left)” – in my opinion, such a comment does not fit the genre of a scientific article.

Thank you for pointing this out. It was a sentence from the first draft that we have failed to remove from the final text. We apologize for the oversight.

Page 6, the Hudson [60] study does not seem, to me, to contribute much to this research. However, the way the authors comment on it is very important: „This study, however, only provides a correlation and doesn’t control for many other variables that might be at play”. I have to say that the four separate models the authors present in the article also fail to „control for many other variables that might be at play”, and, as I will return to it in due course, this conscious choice is still not defended strongly enough.

On the first point, we affirm that Hudson (1963) contributes to the early stages of conceptualization of our study. One of our brainstormed questions could be worded as, Is there an association between personality and aptitude for sciences or humanities? Hudson’s study was one of the earliest attempts at examining this association. Despite its shortcomings, the findings obtained from the study provided an interesting theorization regarding personality types, which we have addressed in our literature review.

On the second point, we would like to point out that the limitations of Hudson (1963) and how they differ from ours. One of the most observable shortcomings is that Hudson’s original dataset only include male pupils, whereas ours include both genders. We have modified the wording of this passage, and we will address our methodological choices as well, in our manuscript as well as later in this letter.

Pages 6 and 7, the fact that the authors review literature on hobbies is commendable, however, seeing that the categories of their own „Hobby” variable does not adopt the categories they find in the literature, and that reading does not feature prominently in the hobby literature either, not much of this section gets used later. One important point being made, however, is that „interest” in a hobby makes the actual practicing of given hobby (very) likely. (This helps the authors' case that their dependent variable is worthy of study.)

While we did not adopt the categories mentioned in the literature review, the conceptualization of pastimes and hobbies in the extant literature have contributed to our own survey and variable design. We also believe that it is important to provide a broad view of the categorizations that have been carried out in past studies and show where our framework fit into the literature. For these reasons, we have decided to keep this section.

Page 7, „In the conceptual framework of this survey, however, we aimed to study the activity of reading in specific relations with education and future occupational aspirations, hence the focus on measurable cognitive and competence-wise benefits of literacy.” Looks very strange in light of the fact that future occupational aspirations are not mentioned in the empirical data / quantitative models. Neither are „measureable cognitive benefits”.

By “this survey” we refer to the full survey, which contains more variables than those mentioned in this singular article. This will further be detailed in the bullet point starting with “Page 9” below, which addresses the sampling methods.

The full list of variables can be found in this article: Vuong, Q. H., Le, A. V., La, V. P., Vuong, T. T., Do, T. H., Vuong, H. M., ... & Ho, M. T. (2019). A Dataset of Vietnamese Junior High School Students’ Reading Preferences and Habits. Data, 4(2), 49. DOI: https://doi.org/10.3390/data4020049

“Future occupational aspirations” are represented mainly by variable “FutureJob”. “Measurable cognitive benefits” refers to variables “APS45” and “APSVNEN”. Explanations for these variables have been provided in the aforementioned article.

Page 7, Section 1.2's last paragraph seems to be inserted largely to satisfy the criticisms of Reviewer 1, but is not integrated enough into the literature review in its present form.

We welcomed the points of critique raised by Reviewer 1 and truly appreciate the accompanying suggestions. As they are mostly tied together and fell under more or less the same argument – the relativity of “reading” as well as print literacy as a measure of cognitive abilities, and the historical implications of the notion of “reading” in light of post-colonialist and post-development views – we found it appropriate to address the topic in one paragraph. We believe it is integrated into the literature review by virtue of its content and its place in the argument. It seems to us that Reviewer 1 did not find our revision of Section 1.2 unsatisfactory.

Page 7, section 1.3, I welcome the clearer deficition of the main dependent variable, and the addition of the reflection on how it might be subject to social desirability.

Thank you very much for the note.

Page 7, section 1.3: „It measure student behaviors regarding the activity, to an extent, and were thus used them as inputs to examine how already formed habits affect the likelihood that they show an interest in reading.” This sentence is the justification of using actual behaviour variables (TimeSci and TimeSoc) as inputs. Although it can be argued that this seems counter-intuitive after the literature review on hobbies, which basically said that, roughly, attitude comes first and behaviour second, also the reverse might be true in many cases, and the authors actually argue their case well here („There is the underlying assumption that people, in their formative years, depend on outside influences to form the habit of engaging in literary activities.”). Perhaps the argumentation might even be strengthened a bit.

Thank you very much for raising this point. We are delighted to hear that our point has come across. We have added certain specifications into the passage, referring to previous sections, to enrich the argument.

Page 7, „Although this measure is also self-reported, it is arguably less under the effect of social desirability due to the fact that junior high students did not seem to interpret this question as reflective of their character.” It does make the reader wonder why the authors are so sure that these items are less under the effect of social desirability, and that „ junior high students did not seem to interpret this question as reflective of their character”.

Thank you for the remark. The main reason for our assumption comes from observing the data. In fact, it appears that students did not hesitate to report spending less than 30 minutes per day reading either type of books, as can be observed in the following distribution table:

TimeSoc

TimeSci

g30

less30

g30

437

174

less30

393

672

Respondents were willing to report the lowest amount of daily reading time for both types of books and less inclined to report not having an interest in reading. We assumed that students do not take the questions “How much time per day do you spend reading natural sciences books?” and “How much time per day do you spend reading social sciences and humanities books?”, which reports their reading behavior, as being potentially reflective of their reading interest. In other words, it seems to us that the students did not make the connection between “TimeSci”, “TimeSoc” and “Readbook”, thus diminishing the effect of social desirability on the responses concerning “TimeSci” and “TimeSoc” relatively to those concerning “Readbook”, hence our statement.

We acknowledge however the fact that the wording of the passage in question might need to be more nuanced. We have modified it accordingly.

Page 8 and onwards, a small but important clarification is needed whether „books on literature/social sciences” includes novels. I.e. books on literature might be scientific works discussing literature; which is not actual literature. An umbrella term „humanities” books might unify literature and social scientific books well enough.

Thank you very much for your input. We have modified the terminology based on your valuable suggestions. As we believe there is a fundamental difference in terms of methodology between the social sciences and humanities disciplines (namely, the philosophical approach and the scientific method), we have decided to use the term “social sciences and humanities” instead of “social sciences and literature”. This should make a clear distinction between the genre in question and books on natural sciences. We have additionally changed “natural science” into plural “natural sciences” for consistency.

These changes were highlighted in yellow. We have also noted the changes in Section 2.1.2. Variables.

Page 8, „we have decided to focus on how the family environment is linked to the reading activity among students.” Does not seem to be a correct claim, „reading activity” might have to be changed to „reading interest”, in accordance with the new name of the dependent variable.

We have changed to “reading interest” in accordance with the new name of the dependent variable.

Page 8, Section 1.3, last paragraph does provide some explanation for the use of binary variables, which I welcome. Still, as will be seen later, the meaning of the Buybook and Readstory variables is not clear in the article. The authors should help the reader in interpreting them; either they are meant in a sense of „regularly occurring”, or more in the sense of „having ever occurred”. The socioeconomic landscape they paint makes the impression that we are talking about something along the lines of the latter.

We thank you for pointing out the potential confusion. We did design the binary variables “Buybook” and “Readstory” to be understood in the sense of “having ever occurred”. We appreciate your suggestion in wording the conceptualization, and have added clarifications in our manuscript correspondingly.

Page 9, sampling is now better explained, which I welcome, but the claims being made do not make it easier for the reader to understand how sampling happened. Both convenience sampling and purposive sampling are, to various degrees, antithetical to „random” sampling, which the authors also claim. My imagination tells me that the data collection was a two-stage sampling process: first selecting schools, then selecting students within schools. Were both sampling steps performed in a random manner? If so, the claim of random sampling can be held. If not, „random” should be dropped from the text, and claims of representativity are also weakened. Also, why this sample of Norther Vietnamese junior high school students can „arguably extend to all junior high students in Vietnam” must be defended. Is it the case that Northern Vietnam represents the whole of the country e.g. with regards to socioeconomic conditions of students? Even if it is the case, the claim that data coming from one region of any country can represent the whole of the country looks very dubious.

We apologize for the lack of precision in the description of our data sampling process as well as the erroneous usage of terminology in the word “random”. To be precise, the dataset used in this manuscript (N=1676) was extracted from the full dataset of over 5000 observations, obtained through our survey. The full dataset contained responses from all students enrolled in all public junior high schools in the province of Ninh Bình; there was no selection in the process. In addition, we do not consider this a two-step sampling process. Rather, the N=1676 dataset in this manuscript is considered the phase 1 result of the survey.

We appreciate your feedback and have added clarifications into the Data section. We have however elected not to include details on the phases of the survey to avoid confusion. Please refer to this article, which we have also cited in concerned sections in our manuscript, for more in-depth data-related details:

Vuong, Q. H., Le, A. V., La, V. P., Vuong, T. T., Do, T. H., Vuong, H. M., ... & Ho, M. T. (2019). A Dataset of Vietnamese Junior High School Students’ Reading Preferences and Habits. Data, 4(2), 49. DOI: https://doi.org/10.3390/data4020049

Page 9, adding a claim such as „In future research studies, different statistical approaches such as Bayesian statistics would be used for data analysis [70]. ” seems unnecessary if the given methods are not used in this very article.

We have moved the claim to the end of the paper as a suggestion for future research: In future research studies, different statistical approaches such as Bayesian statistics could be used for data analysis to test the previous frequentist results [87].

Page 10 now offers a reflection on the fact that the authors made 4 separate models, but this reflection is not a strong justification of the methodological choice. By running 4 separate models, they allow themselves to make the mistake of not controlling „for many other variables that might be at play” in each of their models. Why is this so?

Our methodological choice was in relations with our research questions and hypotheses, each of which concern specific pairs of variables that are related to each other. Namely: “Grade” and “RankinF” are linked to age; “TimeSci” and “TimeSoc” both measure the time spent reading books; “Buybook” and “Readbook” concern the behavior of parents in relation to the child’s book reading; and the ample literature between differing interests and gender has prompted the examination of “Readbook” against “Sex” and “Hobby”. As we purposely wish to explore these associations, we have elected to examine them separately rather than in one large regression model.

We believe one encompassing regression model in which all variables in the current article would complicate the analyses, especially when ordinal variables such as “RankinF” and “Grade” are fitted in the same model as nominal variables. We have the run models in sections 3.2.2 and 3.2.4 with “Sex” included as a variable, all of which yield similar results as presented in the study. In addition, a multiple regression of “Readbook” against “TimeSoc”, “TimeSci”, “Buybook”, “Readstory” and “Sex” has been performed.

The beta coefficients of “Sex” at “male” in the Readbook~Buybook+Readstory+Sex model is -1.03; in the Readbook~TimeSoc+TimeSci+Sex model is -1.07; and in the Readbook~Buybook+Readstory+TimeSoc+TimeSci+Sex model is -1.02 (p-value<0.0001 in all cases). In other words, in all of the aforementioned regressions, “Sex” at “male” has a negative coefficient. This is consistent with the findings reported in Section 3.2.3, that male students are consistently less likely than female students to report positive reading interest (“Readbook” = “yes”). For this reason, we have chosen not to include these models in our manuscript.

Page 10, using the language „nearly half” and „611 out of 1676” in the same sentence is not fortunate. Proportions should be stated in both cases, not frequencies. Also, the last sentence above Table 1 still contains the „had ever read books for them” type of interpretation for the Readstory variable.

We have edited the language and the interpretation.

Page 10, if every independent variable is described in Table 1, I still don't see why 7 more rows for RankinF could not be added. Also, as has been indicated in my first review, there is a problem with mentioning the fact that the highest number of children in a family was 8: the researchers still seem to possess more data than they use. Of course, they are allowed to define the boundaries of their model, but once they let it slip they should justify this.

We apologize for the confusion. We have modified the sentences once again, to now say: “The data shows that “RankinF” ranges from 1 to 7; the largest total number of children in a family is reported to be 8. This means that the highest birth rank reported by our respondents is 7. In other words, none of the respondents is the youngest child in their family.”

What we mean by this is that, while the highest number of children in a family was 8, not all respondents have to necessarily be the last child in their family. In fact, as our data shows, the highest birth rank is 7. This means, if we consider all respondents whose families have 8 children, we find that none of them are the youngest child, but only the second-to-last child, which makes the highest birth rank 7.

For your information, please find below a histogram of variable “RankinF” and the corresponding distribution table (N=1676).

Figure 1. Histogram of variable “RankinF”

RankinF

1

2

3

4

5

6

7

N

695

617

251

79

26

5

3

Table 1. Distribution table of variable “RankinF”

We hope it is now clear enough.

Page 10 still contains references to Appendices, which I could not find in the uploaded manuscript.

We apologize for the oversight. The manuscript now includes the corresponding appendices:

Appendix A: Cross-tabulation of “TimeSoc”, “TimeSci” and “Grade”

Appendix B: Tables of detail probabilities for each research question and hypothesis

Appendix C: Estimate results of “Readbook” by “Grade” and “Hobby”

Page 11 (and elsewhere), let me say this time that Table 3 and its counterparts (Table 5, 7, 10) are overly pedantic. Graphs do a much better job of showing the particular probabilities for the reader. These tables should perhaps be relegated to an Appendix. Also, the equation at the very top of page 12 as an example (of how to calculate a particular probability) is fine once, but should not be repeated below Eq4 on page 17.

We appreciate your input. We have relegated the tables and equations to appendices accordingly.

Page 13 (top) „This aligns with the findings in literature: reading practices and enjoyment decline with age. It appears that Vietnamese students are not exempt from the general global tendency.” perhaps belongs more in the discussion, but this is not a serious issue.

Thank you for the feedback. We do agree that this might also suit the discussion; however, we believe it has a place in the section, both by virtue of its content and its contribution to the articulation of the argument. Therefore, we have decided to keep the passage unchanged.

Page 13, Section 3.2.2. „how stark is the correlation”: this might not be the most felicitous woring (actually it occurs elsewhere too). „Stark” might be fine, although this reviewer has not seen it often in the context of correlations, but seeing that technically, the article does not compute correlations either, the use of „relationship” might be preferable.

We have changed ‘the correlation’ to ‘the relationship’.

Page 13, Table 4. „Grade” and „gr7” are obviously erroneous headings in the table.

We have edited the erroneous headings.

Page 14, „represents students whose parents often read books for them” is another addition to the nebulous interpretation of the variable (now going more in the direction of often instead of ever having had).

Thank you for pointing out the inconsistent wording. We have modified our manuscript accordingly.

Page 15, Table 6: instead of codes a to f, short names for hobbies might help the reader (e.g. „TV/music”, „chores”, „socialize”, „other”). It is interesting that for both RQ2 and RQ3, special attention (i.e. an equation/calculation) is given as to why the null hypothesis is rejected (a better term than „refuted”), but not in the case of RQ1 and RQ4.

On the first point, we have provided a note in the Figure caption to clarify the codes for the reader: Note: a = reading; b = Watching TV/Music; c = Helping with chores; d = Observing nature; e = Socializing; f = Others.

On the second point, we acknowledge that the explicit calculations of the p-value and the rejection of the null hypothesis seems rather redundant. These passages in fact belong to a previous draft of the manuscript, which should have been removed from the final text. We apologize for this oversight and thank you for the input.

Page 15, Section 3.2.3 the sentence beginning with „This is consistent with the extant literature” might belong to the discussion (again not a serious issue).

Thank you for the feedback. Once again we agree that the sentence opens a discussion rather than purely examining the results. That being said, we believe it would be more convenient for the reader to follow the flow of the argument if the point was raised directly in the section and later reprised in the discussion.

Page 16, below figure 3 „In other words, considering reading as a favorite hobby is the strongest predictor to having frequent reading habits” again goes with the old naming of the dependent variable, which is now supposed to be reading interest.

We have changed all the old naming of the dependent variable to the new one: reading interest.

Page 18, seemingly the old name of the dependent variable again, as in „in order to develop reading habits of children”.

We have changed all the old naming of the dependent variable to the new one: reading interest.

Page 19, „There could be, in fact, as many reasons for pupils to lose their taste for reading as there are for them to be interested in reading.” This is certainly not the most worthwhile sentence of the article.

We have removed the sentence.

Page 19, The sentence „Finally, the decline in interest for reading could be a result of pupils’ priorities changing with age. Section 4.3, which deals with, among other aspects, the pastimes of pupils, would examine this in more details.” actually highlights the necessity of a more complete model even more. The authors should have made a model which controls for student age and favourite pastime together. Or, alternatively, to support the quoted claim, they should take a look into whether the hobbies of students do actually change as they get older. They do have the data, after all.

We appreciate your suggestion. In the passage in question, what we meant by “pupils’ priorities” concerns their school-related obligations rather than their hobbies. We acknowledge that the sentence referring to Section 4.3 does not come across as pertinent to the passage and have thus removed it.

Nevertheless, we have included an appendix in which a general linear model has been employed to explore the relationship between student’s grade and favorite pastimes. The results are as follows.

Intercept

Hobby

“Grade”

b

c

d

e

f

logit(yes|no)

7.109***

[9.116]

-0.376***

[-4.692]

-2.306***

[-4.471]

-1.457*

[-2.453]

-1.919**

[-2.783]

-2.445***

[-4.242]

-2.373***

[-4.362]

Significance codes:  0 ‘***’ 0.001 ‘**’ 0.01 ‘*’ 0.05; z-value in [square brackets]; baseline category for: “Grade” = “6”; “Hobby” = “a”. Null deviance: 1073.77  on 1675  degrees of freedom

Residual deviance:  992.26  on 1669  degrees of freedom

AIC: 1006.3

The coefficient  shows that pupils’ reading interest does decline with age (indicated by school grades), regardless of favorite pastime.

Page 19, basically the same issue: „In other words, younger siblings in families with more children receive less attention from parents and thus are less likely to cultivate reading habits since childhood.” The hypothesis that they receive less attention is fine, but there are actually two variables in the model that might stand in for either parental attention and/or resources: Buybook and Readstory. Why are they not checked in the same model with RankinF? Or, again, alternatively: the authors do have the possibility to check whether children with lower rank do receive less parental attention (in the form of parents being less likely to read books for them).

We have considered these possible models. The issue with them is that there are few respondents whose parents buy books for them and even fewer whose parents read stories for them, for reasons related to the socioeconomic landscape of Vietnam, as has been explained in our previous letter. There are also fewer respondents of higher birth ranks, as show in the above histogram of “RankinF”, so the pool of students of higher birth ranks is much smaller than that of students of older birth ranks, which skews the statistical analysis. As a result, even when birth ranks were collapsed into two large categories (“elder”, consisting of “1” through “3”, and “younger”, comprising the rest of the levels), the model yields a coefficient of “RankinF” that is not statistically significant. This does not diminish the possibility that children with higher birth ranks (lower birth ranks denote older children; e.g. “RankinF” = “1” means that the respondent is the oldest child in the family) do receive less parental attention. This simply means that perhaps the variables “Buybook” and “Readstory” are not well suited for measuring parental attention in general, as they were not designed to do so.

Page 19, if „Studies have shown that the incremental effect of parent-child storybook reading diminishes as the child grows older, even in a short timespan, such as from 2- 3 years old to 4-5 years old”, then the fact that the study included Readstory as an independent variable looks strange. If parental book reading has a diminishing effect already by age 5, why did they think it was important for pupils of ages 11-15?

First, the literature on parent reading is not particularly rich and mainly concerns a younger population, namely preschoolers and kindergarteners, as exemplified in the study in question (reference number 78). We don’t consider this a reason to dissuade us from examining the effect of parent reading on older pupils, but a gap in the literature that would be interesting to explore. What if the effect of parental book reading increases when the activity is carried out even as the children grew older? is, for example, one of the questions we posed, during the early stages of the study. We believe it is reasonable not to exclude this possibility. Second, none of the extant literature on this subject is based in an emergent country, regardless of the age of students. We consider this as well to be a gap of the literature, that we might be able to contribute to filling.

As the beta coefficients resulting from our regressions show, parental book reading does still have an observable effect on the reading interest of the respondents. For this reason, we decided not to exclude this variable from the analyses and final manuscript.

Page 20, Section 4.3: the last sentence seems superfluous.

We have removed the sentence.

Page 20, Section 4.4 „our model does not compare the influence of time spent on science books versus that on literary books” – actually it does. The two different beta coefficients obtained do show this differential in influence.

We apologize for the imprecise wording. By this, we mean that the separate influence of TimeSoc or TimeSci on Readbook was not examined in this model, as there are no events in which it is explicitly reported that the student reads exclusively one type of book. (This is because the categories of “TimeSoc” and “TimeSci” do not include an explicit “Never”, rather only offering “Less than 30 minutes per day”.

In other words, the influence of reading natural sciences books and that of reading social sciences and humanities books were analyzed in conjunction. It is true that we compare the differing influence of each type of book in enhancing reading interest. We have reviewed the passage in question in Section 4.4, in order to reflect this idea more clearly. We hope the modifications are satisfactory and welcome any further suggestions.

Page 21, Section 4.5, the first paragraph is a welcome addition to the argument.

Thank you very much for the feedback.

We believe that we have put an appropriate amount of efforts in responding to all of your concerns. We hope that you find our revision satisfactory, and would like to once again thank you for your suggestions. We are truly honored to be able to work with you in improving this manuscript.

With our highest respects,

The authors

Round 3

Reviewer 2 Report

Dear Authors, once more my suggestions are described in the attached document.

Author Response

Dear reviewer,

We would like to address your review with our utmost gratitude, for providing us comprehensive and highly detailed reviews that, we believe, have immensely improved the quality of our submission, not only language-wise but also in the essence of our content. In this round, we sincerely appreciate the time and efforts you have put into helping us point out and correct linguistic or editing errors.

Similarly to the last two rounds, modification of existing passages have been highlighted in yellow, whereas additions are under green highlights.

Following the very clear list compiled in your review, we have revised the entirety of our manuscript and made the changes accordingly. Forgive us for not bringing up each of the points raised; we believe this global passage would be sufficient, as the errors are of similar nature.

We would like to proceed to addressing 4 points that merit individual attention.

Page 2: The illustration.

We thank you for your understanding and acceptance. We would rather not include a caption. Our hope is that the readers would interpret the connection between the illustration and the content of the article, as well as the contribution of the artwork to the insights presented in our study, in their own way.

Page 5: […] It might also help if „reports” were clarified, I believe we are talking about „scientific reports” here.

The original article used the term “reports” with no further clarifications. We have thus opted to use the same terms, to avoid misrepresenting the authors’ ideas. We do agree that they were perhaps referring to “scientific reports”, but we would not dare to assume.

Page 9, even though I have now read the authors' response to me, sampling is still not crystal clear. They state that the full sample of the original survey „concerns all adolescent students” of the province. If this is really the case, then the original survey made a kind of „complete” sample, which is actually a very good thing, and if that is really the case, labels such as „purposive” and „convenience” should not even apply. However, now that it is clear that the original survey has a sample of some 5000 students, and even the article makes the claim that the 1676 observations are „a subset of the data”, the authors should provide some explanation as to why and how they arrived at this subsample. Are these 1676 instances a random selection from the total dataset? I would welcome some further clarification of this matter here.

We confirm that the full dataset is indeed a complete dataset of the province of Ninh Binh; what we meant by purposive and convenient was only in regards to representativity on a national scale. We have rephrased the passage with clearer references to the original survey as well as the sampling methods, with references to the cited data article. We have also mentioned that the subset was compiled randomly in the data-entering process.

Page 22, the sentence about Bayesian statistics is now very out of context within the paragraph where it is placed. Perhaps the authors should re-read their article and collect all of their remarks about possibilities of future research and group them together in a separate paragraph at the end of their article.

We have revised and rewritten the passage to better articulate our idea. We hope that the reference is now integrated into the flow of our argument.

Once again, we hope that our manuscript has been improved in a satisfactory manner. We sincerely appreciate your suggestions and propositions in order to help us reinforce the scientific soundness of the paper, as well as smoothing out the flow of the writing and the quality of the presentation. We welcome all remarks and feedback that you have dedicated your time into making. It is a truly enriching experience to work with you.

With our highest respects,

The authors